# When Active Learning Meets Graph Similarity: Evidential Variance for Graph Selection

**Chengtai Cao**                                                    *chengtcao2-c@my.cityu.edu.hk*
*Department of Computer Science, City University of Hong Kong*

**Haoyu Yang**                                                           *yhy818@std.uestc.edu.cn*
*School of Information and Software Engineering, University of Electronic Science and Technology of China*

**Shenglin Wang**                                                     *swang586-c@my.cityu.edu.hk*
*Department of Computer Science, City University of Hong Kong*

**Xinglin Lian**                                                    *kenshin.lian24@gmail.com*
*School of Information and Software Engineering, University of Electronic Science and Technology of China*

**Fan Zhou**[*]                                                          *fan.zhou@uestc.edu.cn*
*School of Information and Software Engineering, University of Electronic Science and Technology of China*
*Intelligent Digital Media Technology Key Laboratory of Sichuan Province*

**Reviewed on OpenReview:** *https://openreview.net/forum?id=dV6UopxOjX*

## Abstract

Graph Similarity Learning (GSL) is pivotal in graph data mining, yet training effective models necessitates substantial labeled pairs, which incur prohibitive annotation costs. To address this, we introduce Active Learning (AL) into the GSL paradigm. However, directly transferring existing AL strategies is non-trivial due to two unique impediments: (1) the *continuous regression nature* of similarity prediction complicates standard uncertainty quantification, and (2) the *paired-input structure* requires evaluating a graph's informational value across its pairings rather than in isolation. To bridge this gap, we propose **EVGS** (Evidential Variance for Graph Selection), a novel AL framework tailored for GSL. EVGS leverages evidential deep learning to impose a prior over predictions, enabling disentangled uncertainty estimation. Crucially, we identify a "gradient shrinkage" pathology inherent to the data-scarce regime characteristic of AL cycles. We introduce a novel MSE-anchored regularizer to mitigate this issue, ensuring discriminative uncertainty estimation even with limited labels. Furthermore, to address the paired-input challenge, we propose a graph-centric selection criterion: *uncertainty variance*. This metric captures a graph's holistic informational value by measuring fluctuations in its epistemic uncertainty across diverse interactions. Extensive experiments on three benchmarks with two GSL backbones demonstrate that EVGS consistently outperforms established AL baselines.

## 1 Introduction

Graph-structured data is ubiquitous in modeling complex relationships across diverse domains, ranging from social networks to chemical molecules (Ju et al., 2025). In these contexts, measuring graph similarity serves as a fundamental task, underpinning critical applications such as graph retrieval, malware detection, and brain network analysis (Ma et al., 2021). While traditional algorithms like Graph Edit Distance (GED) (Gao et al., 2010) provide exact metrics, they are known to be NP-complete (Bunke & Shearer, 1998), rendering them computationally prohibitive for large-scale real-world datasets.

---

[*]Corresponding author

To circumvent this computational bottleneck, recent research has pivoted towards **Graph Similarity Learning (GSL)**, which leverages Graph Neural Networks (GNNs) (Wu et al., 2020) to efficiently approximate similarity metrics (Li et al., 2019; Wang et al., 2021; Ranjan et al., 2022; Tan et al., 2023; Lan et al., 2024; Zheng et al., 2025; Zou et al., 2025). By training on annotated graph pairs, GSL models, including both embedding-based (Bai et al., 2019; Zhang et al., 2021d) and interaction-based architectures (Ling et al., 2019; Xu et al., 2021), enable rapid similarity inference on unseen graph pairs.

Despite their promising performance, a critical limitation persists: these data-driven models are notoriously data-hungry (Li et al., 2018). Obtaining the necessary ground-truth labels (e.g., exact GED values) requires running expensive algorithms on massive datasets, creating a paradoxical situation where training an efficient model incurs a prohibitive initial cost (Cai et al., 2017). For example, training a standard GSL model such as SimGNN (Bai et al., 2019) often demands millions of labeled pairs, imposing a severe annotation burden (Sorokin & Forsyth, 2008). Consequently, **Active Learning (AL)** (Settles, 2009) emerges as a vital solution, offering a principled framework to strategically select only the most informative samples for annotation, thereby maximizing performance under limited budgets.

However, adapting AL strategies from general machine learning (Settles, 2009; Ren et al., 2021) to the GSL domain is non-trivial due to a fundamental task mismatch. Existing graph AL methods predominantly focus on node classification (Gao et al., 2018; Zhang et al., 2021a; 2022; Sheng et al., 2025; Chen et al., 2026), typically selecting nodes based on prediction entropy. In contrast, GSL operates on graph pairs for similarity regression, presenting two inherent challenges that render off-the-shelf strategies ineffective:

- **Challenge 1: Uncertainty Quantification in Regression.** Unlike classification, where entropy naturally measures uncertainty over discrete classes, GSL outputs continuous similarity scores. Consequently, standard entropy-based metrics are inapplicable, necessitating alternative uncertainty measures that are compatible with the continuous output space of regression models.

- **Challenge 2: Structural Dependency in Paired Inputs.** GSL inputs are combinatorial pairs, yet the underlying information stems from individual graphs. Naive pair-level selection ignores this dependency: it treats each pair as an isolated instance, overlooking that a single graph participates in multiple relationships. Simply evaluating uncertainty at the pair level is myopic, failing to capture the graph's holistic information value for similarity-space learning. Therefore, effective selection must account for each graph's collective impact across its associated pairs.

To address these challenges, we propose **E**vidential **V**ariance for **G**raph **S**election (**EVGS**), a novel and general AL framework designed to identify the most informative graphs for GSL. EVGS tackles the aforementioned obstacles through two key innovations. First, to quantify regression uncertainty (Challenge 1), we adapt **evidential deep learning** (Sensoy et al., 2018; Amini et al., 2020) to place a higher-order prior over model predictions, enabling the joint estimation of aleatoric and epistemic uncertainty. Crucially, we identify a "gradient shrinkage" pathology in standard evidential learning within low-data regimes and introduce a novel **MSE-anchored regularizer**. This mechanism mitigates vanishing gradients, ensuring discriminative and reliable uncertainty estimation, a prerequisite for robust active selection. Second, to resolve the combinatorial dependency (Challenge 2), we introduce a new selection criterion: **uncertainty variance**. Instead of evaluating pairs in isolation, this metric computes the variance of a candidate graph's epistemic uncertainty across its interactions with other graphs. By prioritizing graphs that exhibit high variability in model confidence across different contexts, EVGS captures the holistic informational value of each graph. Importantly, EVGS is model-agnostic, seamlessly integrating with various GSL architectures to provide a versatile solution for data-efficient GSL. In summary, our main contributions are threefold:

- **Pioneering AL for GSL:** To the best of our knowledge, this is the first work to formalize AL specifically for GSL. We identify the unique challenges of applying AL to paired-input regression tasks, bridging the critical gap between data-efficient learning and graph matching.

- **Methodological Innovation:** We propose EVGS, a novel framework that synergizes evidential deep learning with a tailored MSE-anchored regularizer to mitigate gradient shrinkage, and introduces a structure-aware uncertainty variance metric to capture holistic informational value.

- **Empirical Validation:** Extensive experiments across three benchmarks and two GSL backbones show that EVGS consistently outperforms established AL baselines. Our results validate both the efficacy of our uncertainty estimation and the necessity of graph-centric selection strategies.

## 2 Related Work

### 2.1 Graph Similarity Learning (GSL)

Traditional approaches to graph similarity, exemplified by Graph Edit Distance (GED) (Gao et al., 2010) and Maximum Common Subgraph (MCS) (Bunke & Shearer, 1998), provide rigorous theoretical foundations but are computationally intractable. Specifically, the exact computation of these metrics is known to be NP-hard (Bunke & Shearer, 1998), rendering them impractical for large-scale real-world scenarios. To circumvent this scalability bottleneck, graph kernels (Yan et al., 2005; Yanardag & Vishwanathan, 2015; Nikolentzos et al., 2017) were introduced as an alternative; however, they often suffer from shallow expressiveness, limiting their generalization capabilities. Consequently, the field has witnessed a paradigm shift towards Graph Neural Networks (GNNs) (Wu et al., 2020; Ju et al., 2025). By learning expressive data-driven representations, GNN-based GSL methods have achieved state-of-the-art performance across diverse applications (Li et al., 2019; Ma et al., 2019; Doan et al., 2021; Qin et al., 2021; Wang et al., 2021; Ranjan et al., 2022).

Early embedding-based approaches typically estimate similarity by comparing independently learned graph-level representations using metrics such as cosine or Hamming distance (Zhang et al., 2021d). To better capture substructural intricacies, SMPNN (Riba et al., 2018) and HGMN (Ling et al., 2019) introduced mechanisms to aggregate node-node similarity scores and model cross-graph interactions, respectively. Further enhancing granularity, SimGNN (Bai et al., 2019) incorporates a learnable neural tensor network with histogram features, while GraphSim (Bai et al., 2020) employs CNNs to process node similarity matrices as images. Subsequent research, including MGMN (Ling et al., 2021) and hierarchical approaches (Xu et al., 2020; 2021), explores multi-level matching and graph partitioning to facilitate effective cross-scale comparisons. Building upon these interaction-focused paradigms, NA-GSL (Tan et al., 2023) proposes a unified framework that leverages cross-graph co-attention and similarity-wise self-attention for precise structural alignment. Most recently, GraSP (Zheng et al., 2025) has advanced the boundaries of expressiveness and efficiency through multi-scale pooling and positional encodings. Parallelly, AMFF (Zou et al., 2025) introduces an adaptive feature fusion mechanism to dynamically adjust weights based on node-graph interactions.

Despite these architectural innovations, a critical bottleneck persists: existing models are predominantly data-hungry, requiring substantial annotated datasets to ensure generalization. This reliance incurs prohibitive annotation costs, especially given the complexity of labeling graph pairs. To address this challenge, we introduce a general Active Learning (AL) framework tailored explicitly for GSL.

### 2.2 Active Learning (AL)

**Common AL.** AL aims to maximize model performance under a limited budget by iteratively querying labels for the most valuable instances. Existing strategies generally fall into three categories: informativeness-based, representativeness-based, and hybrid approaches. Informativeness-based methods prioritize instances where the model is highly uncertain. Beyond classic techniques such as Uncertainty Sampling (Settles, 2009) and Query-by-Committee (QBC) (Seung et al., 1992), recent advances, such as Learning Loss (Yoo & Kweon, 2019), extend this paradigm to deep learning by training a module to predict the target loss of unlabeled samples. Representativeness-based approaches ensure the selected subset covers the underlying data distribution. While traditional methods rely on density (Xu et al., 2007) or clustering metrics (Nguyen & Smeulders, 2004), the influential Coreset (Sener & Savarese, 2018) algorithm adapts this to deep networks by formulating sample selection as a $k$-Center problem in the feature space. Hybrid strategies synergize these objectives to achieve a balance between uncertainty and diversity. For example, IR-AL (Yang et al., 2015) explicitly maximizes diversity within uncertainty sampling, while BADGE (Ash et al., 2020) selects samples with high gradient magnitudes (uncertainty indicator) that are simultaneously diverse in the gradient space.

**AL on Graph.** AL on graphs presents unique challenges, necessitating strategies that explicitly account for topological dependencies and relational complexities (Yang et al., 2025a;b). Initial efforts, such as AGE (Cai et al., 2017) and ANRMAB (Gao et al., 2018), adapted traditional criteria by incorporating node centrality and embedding density. Recognizing the importance of connectivity, subsequent methods like ALG (Zhang et al., 2021a) and Grain (Zhang et al., 2021c) focus on influence maximization, prioritizing nodes that effectively propagate information to their neighbors. Beyond structural heuristics, researchers have explored more sophisticated selection mechanisms. For example, SEAL (Li et al., 2020) and GALE (Guan et al., 2023) employ adversarial strategies to probe graph structures, while IGP (Zhang et al., 2022) introduces soft-labeling to mitigate annotation ambiguity. In parallel, several works address practical deployment constraints: RIM (Zhang et al., 2021b) tackles noisy oracles, and ALLIE (Cui et al., 2022) handles large-scale class imbalance. Most recently, FSV (Chen et al., 2026) offers a novel perspective by tracking feature dynamics, while DMA (Sheng et al., 2025) integrates dataset-specific characteristics with LLM capabilities.

Despite these advancements, a critical gap remains: existing graph AL methods mainly focus on *node classification*. In contrast, GSL operates on *graph pairs* for *similarity regression*. This fundamental divergence presents two inherent challenges: (1) the continuous nature of regression targets complicates uncertainty quantification, and (2) the combinatorial structure of paired inputs requires holistically evaluating a graph's value. To this end, we introduce EVGS, a general framework tailored for the paired-input regression task.

## 3 Preliminaries

### 3.1 Graph Similarity Learning

**Definition 1** *Graph Similarity Learning (GSL). Let $\mathcal{G}$ be the space of all graphs. Given a pair of graphs $(G, G') \in \mathcal{G} \times \mathcal{G}$, GSL aims to learn a regression function $f : \mathcal{G} \times \mathcal{G} \to \mathbb{R}$ that predicts a similarity score $\hat{y}$ approximating the ground-truth metric $y$.*

Current GSL architectures typically employ Siamese GNNs. Given a pair $(G, G')$, a weight-sharing GNN encoder first extracts node representations $\mathbf{H}$ and $\mathbf{H}'$. A readout function then aggregates these into graph embeddings $\mathbf{g}$ and $\mathbf{g}'$. To capture the interaction between these two graphs, a fusion module $\phi$ constructs a graph-pair representation $\mathbf{z}$, which is subsequently mapped to the scalar similarity score by a regressor $\psi$:

$$\mathbf{z} = \phi(\mathbf{g}, \mathbf{g}', \mathbf{H}, \mathbf{H}'), \quad \hat{y} = \psi(\mathbf{z}). \tag{1}$$

While effective, training $\phi$ and $\psi$ modules is data-hungry. Given the high cost of ground-truth annotation for GSL, we propose integrating AL to achieve high performance with minimal labeled data.

### 3.2 Pool-based Batch-mode Active Learning

In this work, we adopt the pool-based batch-mode active learning paradigm (Sugiyama & Nakajima, 2009; Cai et al., 2016) to maximize the sample efficiency of the GSL model $f$.

**Problem Setup.** The active learning process operates on the space of graph pairs, initialized with a small seed set $\mathbb{L}_0$ and a large unlabeled pool $\mathbb{U}_0 \subseteq \mathcal{G}_{\text{pool}} \times \mathcal{G}_{\text{pool}}$, where $\mathcal{G}_{\text{pool}}$ denotes the collection of available graphs. The learning procedure spans $T$ rounds. In each round $t$, the GSL model $f$ is trained on $\mathbb{L}_{t-1}$. Subsequently, a query strategy selects a batch $\Delta_t$ of informative pairs from $\mathbb{U}_{t-1}$ for annotation. The ground-truth similarity scores for $\Delta_t$ are queried from an Oracle (e.g., an exact GED solver), which is typically computationally intensive. The datasets are then updated as $\mathbb{L}_t \leftarrow \mathbb{L}_{t-1} \cup \Delta_t$ and $\mathbb{U}_t \leftarrow \mathbb{U}_{t-1} \setminus \Delta_t$.

**Query Level: Pairs vs. Graphs.** A fundamental challenge in active GSL arises from the combinatorial nature of the input space. While the GSL model and the Oracle operate on pairs, the underlying dataset is composed of individual graphs. This discrepancy necessitates a strategic choice regarding the selection level:

- Pair-level Selection: Treats the unlabeled pool as a flat set of independent pairs, directly selecting the most informative instances $(G^i, G^j)$ to form the query pair batch $\Delta_t$.

- Graph-centric Selection: Shifts the focus to individual graphs. Unlike direct pair sampling, this entails a *constructive process*: it first identifies a batch of critical graphs $\mathcal{G}_t \subseteq \mathcal{G}_{\text{pool}}$, and subsequently generates the query pair batch $\Delta_t$ by forming all pairwise constraints within $\mathcal{G}_t$. Finally, the candidate graph pool is updated: $\mathcal{G}_{\text{pool}} \leftarrow \mathcal{G}_{\text{pool}} \setminus \mathcal{G}_t$.

For both strategies, we define the annotation budget strictly in terms of the number of queried pairs per round. To maintain consistency across different selection levels, we align the number of selected graphs $m$ with the pair budget $b = |\Delta_t|$. Specifically, under the graph-centric strategy where the query batch is constructed from all pairwise combinations within the selected graph set, the relationship is governed by $b = \binom{m}{2}$. This formulation allows us to derive the required graph batch size $m$ for a given pair budget $b$. For example, selecting $m = 25$ graphs corresponds to an annotation cost of $b = 300$ pairs.

In this work, we advocate for the **graph-centric** strategy. We argue that standard pair-level selection is suboptimal because it treats similarity learning as isolated comparisons, neglecting the fact that these pairs share common operands. Specifically, pair-level methods are prone to the "hubness" trap: they tend to repeatedly query pairs involving the same ambiguous graph (e.g., $\{(G^a, G^1), (G^a, G^2), \dots\}$) simply because $G^a$ itself is hard to characterize. This results in information redundancy concentrated on narrow local regions. In contrast, the graph-centric approach treats selected graphs as anchors. By systematically evaluating an anchor against the population, we acquire a holistic profile of its position in the metric space, rather than merely resolving individual ambiguities. This fosters a more efficient coverage of the global similarity landscape. Empirical validation of this hypothesis is provided in Section 5.2.1 and Section 5.3.2.

## 4 Methodology

This section details the EVGS framework, which integrates robust uncertainty quantification with graph-centric querying. We propose a regularized evidential regression mechanism enhanced with a gradient-anchoring term to ensure reliable uncertainty estimation. Based on this, we leverage the estimated uncertainty to select informative graphs, thereby maximizing information gain within the learned metric space.

### 4.1 Uncertainty Estimation via Regularized Evidential Regression

#### 4.1.1 Probabilistic Modeling via Evidential Regression

**Evidential Regression.** Standard GSL models typically output deterministic similarity scores, thereby failing to capture the uncertainty estimates crucial for AL. While Bayesian approaches like MC-Dropout (Gal & Ghahramani, 2016) or Deep Ensembles (Lakshminarayanan et al., 2017) can provide such estimates, they often incur high computational costs or memory overheads and yield only coarse approximations. To achieve both reliability and efficiency, we adopt Evidential Regression (Amini et al., 2020; Lian et al., 2025; 2026). This framework enables uncertainty quantification within a single forward pass by modeling the target as the hyperparameters of a higher-order evidential distribution, which supports uncertainty decomposition.

Formally, rather than modeling the generative process of the graphs themselves, we treat the continuous scalar similarity score $y \in \mathbb{R}$ between a graph pair $(G, G')$ as a sample drawn from a Gaussian distribution with an unknown mean $\mu$ and variance $\sigma^2$. To capture the uncertainty governing these parameters, we place a Normal-Inverse-Gamma (NIG) conjugate prior over them (Wu et al., 2024; Ye et al., 2024):

$$y \sim \mathcal{N}(\mu, \sigma^2), \quad (\mu, \sigma^2) \sim \text{NIG}(\gamma, \nu, \alpha, \beta) \triangleq \mathcal{N}\left(\mu \mid \gamma, \frac{\sigma^2}{\nu}\right) \Gamma^{-1}(\sigma^2 \mid \alpha, \beta), \tag{2}$$

where $\Gamma^{-1}$ denotes the inverse-gamma distribution. In this formulation, $\gamma \in \mathbb{R}$ represents the estimated similarity score, $\nu > 0$ quantifies the "predicted evidence" supporting the mean, while $\alpha > 1$ and $\beta > 0$ govern the scale and shape of the variance distribution.

To implement this, we introduce a minimal modification to the GSL backbone. Specifically, we replace the final projection layer with a prediction head that outputs the four hyperparameters $\mathbf{o} = \{\gamma, \nu, \alpha, \beta\}$. This design incurs negligible computational overhead while enabling comprehensive uncertainty quantification.

**Uncertainty Decomposition.** A key advantage of this framework is the analytical decomposition of uncertainty. Based on the predicted moments of the NIG distribution, we can derive the prediction $\hat{y}$, aleatoric uncertainty $\mathcal{U}_{\text{ale}}$, and epistemic uncertainty $\mathcal{U}_{\text{epi}}$ as follows:

$$\underbrace{\hat{y} = \mathbb{E}[\mu] = \gamma}_{\text{Prediction}}, \quad \underbrace{\mathcal{U}_{\text{ale}} = \mathbb{E}[\sigma^2] = \frac{\beta}{\alpha - 1}}_{\text{Aleatoric Uncertainty}}, \quad \underbrace{\mathcal{U}_{\text{epi}} = \text{Var}[\mu] = \frac{\beta}{\nu(\alpha - 1)}}_{\text{Epistemic Uncertainty}}. \tag{3}$$

We provide the detailed step-by-step mathematical derivation of these uncertainty components in Appendix A.1. Crucially, this decomposition disentangles data-inherent noise ($\mathcal{U}_{\text{ale}}$) from model ignorance ($\mathcal{U}_{\text{epi}}$). In the context of AL, this distinction is paramount. Aleatoric uncertainty arises from the natural complexity or noise within the data distribution and is theoretically irreducible. In contrast, epistemic uncertainty reflects the model's lack of knowledge due to data scarcity. Since the objective of AL is to reduce ignorance, querying instances dominated by high aleatoric uncertainty is inefficient (Park et al., 2023). Accordingly, we formally define our uncertainty metric exclusively based on the epistemic component $\mathcal{U}_{\text{epi}}$.

### 4.1.2 Optimization Objective and the Proposed Regularization

**The Standard Evidential Objective.** Following the evidential regression framework (Amini et al., 2020), we optimize the model by maximizing the marginal likelihood. Let $\mathbf{o} = \{\gamma, \nu, \alpha, \beta\}$ denote the predicted evidential parameters for a given graph pair with ground-truth similarity $y$. The marginal likelihood $p(y|\mathbf{o})$ is obtained by integrating out the latent Gaussian parameters $(\mu, \sigma^2)$:

$$p(y|\mathbf{o}) = \int_{\sigma^2=0}^{\sigma^2=\infty} \int_{\mu=-\infty}^{\mu=\infty} p\left(y|\mu, \sigma^2\right) p\left(\mu, \sigma^2|\mathbf{o}\right) d\mu d\sigma^2 = \text{St}\left(y; \gamma, \frac{\beta(1+\nu)}{\nu\alpha}, 2\alpha\right). \tag{4}$$

The complete derivation of this marginalization is detailed in Appendix A.2. This integration yields a Student-t distribution $\text{St}(\cdot)$ with location $\gamma$, scale $\frac{\beta(1+\nu)}{\nu\alpha}$, and $2\alpha$ degrees of freedom. Consequently, the primary learning objective is to minimize the Negative Log-Likelihood (NLL):

$$\mathcal{L}_{\text{NLL}} = -\log(p(y|\mathbf{o})) = \frac{1}{2}\log\left(\frac{\pi}{\nu}\right) - \alpha\log(\Omega) + \left(\alpha + \frac{1}{2}\right)\log\left((y-\gamma)^2\nu + \Omega\right) + \log\left(\frac{\Gamma(\alpha)}{\Gamma\left(\alpha + \frac{1}{2}\right)}\right), \tag{5}$$

where $\Gamma(\cdot)$ is the Gamma function and $\Omega = 2\beta(1+\nu)$.

In standard evidential regression (Amini et al., 2020), this objective is typically augmented with a heuristic regularizer, $\mathcal{L}_{\text{REG}}$, designed to constrain the evidence accumulation process:

$$\mathcal{L}_{\text{REG}} = |y - \gamma| \cdot (2\nu + \alpha). \tag{6}$$

Thus, the base objective is formulated as $\mathcal{L}_{\text{base}} = \mathcal{L}_{\text{NLL}} + \lambda\mathcal{L}_{\text{REG}}$, where $\lambda$ is a balancing coefficient.

**The Low-Evidence Trap in Active Learning.** While theoretically sound, the evidential objective exhibits a critical pathology, particularly during the early stages of AL when labeled data is scarce. Let us examine the gradient of the dominant term, $\mathcal{L}_{\text{NLL}}$, with respect to the prediction $\gamma$:

$$\frac{\partial\mathcal{L}_{\text{NLL}}}{\partial\gamma} = \frac{(2\alpha+1)(\gamma-y)\nu}{(y-\gamma)^2\nu + 2\beta(1+\nu)}. \tag{7}$$

We observe that the gradient magnitude is proportional to the evidence $\nu$. In the low-data regime characteristic of early AL cycles, the model tends to predict low evidence (i.e., $\nu \to 0^+$). Consequently, the gradient $\frac{\partial\mathcal{L}_{\text{NLL}}}{\partial\gamma}$ vanishes, regardless of the magnitude of the prediction error. This creates a generic optimization trap: the model can trivially minimize the NLL by simply predicting zero evidence for all samples.

This phenomenon is catastrophic for our AL framework. The query strategy in EVGS relies on relative differences in epistemic uncertainty $\mathcal{U}_{\text{epi}}$ (which is inversely proportional to $\nu$) to distinguish valuable instances.

When the model falls into this low-evidence trap, it assigns uniformly high uncertainty to the entire unlabeled pool. This flattens the acquisition landscape, rendering the selection strategy effectively equivalent to random sampling and hindering the model's iterative improvement.

One might expect the standard regularizer, $\mathcal{L}_{\text{REG}}$, to alleviate this issue by providing auxiliary gradients. However, this mechanism is fundamentally misaligned with AL's needs in the low-data regime. First, the gradient signal from $\mathcal{L}_{\text{REG}}$ scales with the evidence magnitude (Oh & Shin, 2022); thus, in the "uniform ignorance" state ($\nu \to 0^+$), the corrective signal effectively vanishes. Second, and more critically, $\mathcal{L}_{\text{REG}}$ is designed to penalize overconfidence on errors (Amini et al., 2020), not to encourage evidence accumulation. Paradoxically, this incentivizes the model to further suppress evidence (reducing $\nu$) to minimize the regularization penalty. Instead of restoring the discriminative uncertainty landscape required for effective sample ranking, standard regularization can inadvertently exacerbate the collapse into a flat, uninformative state.

**MSE as a Gradient Anchor for Uncertainty Estimation.** To counteract the vanishing gradient problem, we explicitly introduce a Mean Squared Error (MSE) term to anchor the optimization:

$$\mathcal{L}_{\text{MSE}} = (y - \gamma)^2. \tag{8}$$

Crucially, the gradient $\frac{\partial \mathcal{L}_{\text{MSE}}}{\partial \gamma} = 2(\gamma - y)$ is independent of the evidence $\nu$. This term serves as a consistent "gradient injection", ensuring that the model receives robust updates for the target parameter even when it falls into the low-evidence trap ($\nu \to 0^+$). This prevents the model from minimizing the NLL objective by simply outputting "uniform ignorance" to mask prediction errors. Consequently, the epistemic uncertainty derived from the evidential parameters is forced to reflect actual data density rather than optimization failures, thereby restoring the discriminative landscape necessary for effective AL queries. Beyond this theoretical analysis, Section 5.2.2 provides empirical evidence demonstrating the low-evidence trap in AL for vanilla evidential regression, alongside the effectiveness of our proposed MSE-based gradient anchor.

Combining these components, the training objective in our EVGS framework, $\mathcal{L}_{\text{TOTAL}}$, is defined as:

$$\mathcal{L}_{\text{TOTAL}} = \mathcal{L}_{\text{NLL}} + \lambda_1 \mathcal{L}_{\text{REG}} + \lambda_2 \mathcal{L}_{\text{MSE}}, \tag{9}$$

where $\lambda_1$ and $\lambda_2$ are hyperparameters balancing the regularization strength and the gradient-anchoring effect, respectively. In this composite framework, $\mathcal{L}_{\text{NLL}}$ drives the learning of the evidential distribution, $\mathcal{L}_{\text{REG}}$ penalizes overconfidence on outliers, and $\mathcal{L}_{\text{MSE}}$ ensures continuous gradient flow to prevent evidence collapse, collectively enabling robust uncertainty estimation for AL.

## 4.2 Graph-Centric Selection via Uncertainty Variance

While conventional AL strategies prioritize instances with maximal epistemic uncertainty, applying this naively to GSL is suboptimal due to the combinatorial nature of the input space. A purely pairwise perspective is structurally myopic: it treats comparisons as isolated events, ignoring that they share common constituent graphs. This often leads to the "hubness" trap, where the model redundantly queries multiple pairs involving the same ambiguous graph, yielding diminishing returns. To overcome this, we adopt a graph-centric strategy. By evaluating how a graph interacts with the population, we capture its holistic position in the metric space rather than merely resolving local ambiguities.

To implement this graph-centric perspective, we propose that the informativeness of a graph $G^i$ is effectively captured by the variability of the model's uncertainty regarding it. Intuitively, if a graph lies in a well-learned region, the model should exhibit consistently low uncertainty across comparisons; conversely, if a graph is a "hard" sample, uncertainty tends to be consistently high. The most valuable instances are those situated in ambiguous regions of the metric space, where the model's epistemic uncertainty fluctuates significantly.

Formally, for each candidate graph $G^i$ in the pool $\mathcal{G}_{\text{pool}}$, we quantify this fluctuation by measuring the variance of its epistemic uncertainty. To circumvent the prohibitive $O(N^2)$ computational cost of exhaustive pairing ($N = |\mathcal{G}_{\text{pool}}|$), we employ a stochastic approximation. Specifically, for each $G^i$, we sample a random reference subset $\mathcal{S}_i \subseteq \mathcal{G}_{\text{pool}}$ of size $K$ ($K \ll N$). This approximation reduces the selection complexity from quadratic to linear $O(NK)$, ensuring scalability for large-scale datasets. The selection score $s^i$ is defined as:

$$s^i = \text{Var}\left\{\mathcal{U}_{\text{epi}}^{ij} \mid G^j \in \mathcal{S}_i\right\}. \tag{10}$$

---

**Algorithm 1:** Evidential Variance for Graph Selection (EVGS)

---

**Input:** Seed labeled set $\mathbb{L}_0$, Unlabeled set $\mathbb{U}_0$, Candidate graph pool $\mathcal{G}_{\text{pool}}$, Graph budget $m$ per cycle, Max active learning cycles $T$, Reference graph set size $K$, Hyperparameters $\lambda_1$ and $\lambda_2$.

**Output:** Trained GSL model $f$.

**1 for** *cycle $t = 1$ to $T$* **do**

    // Phase 1: Robust GSL Backbone Training

**2**    Train $f$ on $\mathbb{L}_{t-1}$ by minimizing the total objective: $\mathcal{L}_{\text{TOTAL}} = \mathcal{L}_{\text{NLL}} + \lambda_1 \mathcal{L}_{\text{REG}} + \lambda_2 \mathcal{L}_{\text{MSE}}$;

    // Phase 2: Variance-based Graph Selection

**3**    Initialize scores $\mathbb{S} = \emptyset$;

**4**    **for** *each candidate graph $G^i \in \mathcal{G}_{pool}$* **do**

**5**        Sample a random reference subset $\mathcal{S}_i \subseteq \mathcal{G}_{\text{pool}}$ of size $K$;

**6**        Compute epistemic uncertainty $\mathcal{U}_{\text{epi}}^{ij}$ for all pairs $\{(G^i, G^j) \mid G^j \in \mathcal{S}_i\}$;

**7**        Calculate selection score via uncertainty variance: $s^i = \text{Var}\left\{\mathcal{U}_{\text{epi}}^{ij} \mid G^j \in \mathcal{S}_i\right\}$;

**8**        $\mathbb{S} \leftarrow \mathbb{S} \cup \{s^i\}$;

**9**    Select $m$ graphs corresponding to the highest scores in $\mathbb{S}$, obtaining $\mathcal{G}_t$;

**10**   Generate the query pair batch $\Delta_t$ by forming all pairwise constraints within $\mathcal{G}_t$;

    // Phase 3: Update

**11**   Query labels for $\Delta_t$ and update sets:

**12**   $\mathbb{L}_t \leftarrow \mathbb{L}_{t-1} \cup \Delta_t, \ \mathbb{U}_t \leftarrow \mathbb{U}_{t-1} \setminus \Delta_t, \ \mathcal{G}_{\text{pool}} \leftarrow \mathcal{G}_{\text{pool}} \setminus \mathcal{G}_t$;

**13 return** $f$

---

We prioritize variance over conventional mean uncertainty to explicitly target relational ambiguity rather than inherent difficulty. A graph with consistently high uncertainty often indicates an instance with unique substructures that the model struggles to match against references. While difficult, such graphs may be plagued by irreducible noise or domain shifts, yielding diminishing returns for the active learner. From a generalization perspective, targeting these high-variance instances stabilizes the global metric structure. Since GSL aims to approximate a smooth similarity manifold, graphs exhibiting erratic confidence fluctuations act as local "singularities" that disrupt this smoothness. Annotating these instances compels the model to resolve such inconsistencies, effectively "pinning" the embedding of $G^i$ to a stable location that satisfies relational constraints. This resolves topological ambiguities and facilitates more effective gradient propagation throughout the similarity network. Empirical validation of this hypothesis is provided in Section 5.2.1.

Putting this into practice, the complete EVGS workflow is outlined in Algorithm 1. The process alternates between optimizing the evidential GSL model with our MSE-anchored objective ($\mathcal{L}_{\text{TOTAL}}$) and selecting instances via the stochastic variance score ($s^i$). This iterative loop ensures that the model stabilizes regions with inconsistent labels while maintaining robust gradient signals throughout the active learning lifecycle.

## 5 Experiments

### 5.1 Experimental Settings

**Datasets.** We evaluate EVGS on three widely recognized benchmarks: AIDS (Liang & Zhao, 2017), LINUX (Wang et al., 2012), and IMDB (Yanardag & Vishwanathan, 2015). These datasets span diverse domains, representing chemical compounds, program dependency graphs, and social interaction networks. Following standard protocols, we adopt a random 60%/20%/20% split for training, validation, and testing.

**Active Learning Protocols.** We initialize the training process by randomly sampling a seed set of graphs $\mathcal{G}_0$ from each dataset. To construct the initial labeled pool $\mathbb{L}_0$, we generate a fully connected graph of constraints, pairing all instances within $\mathcal{G}_0$. Consequently, the initial labeled set size is $|\mathbb{L}_0| = \binom{|\mathcal{G}_0|}{2}$. Specifically, we set $|\mathcal{G}_0|$ to 70, 100, and 150 for AIDS, LINUX, and IMDB, respectively. This results in 2,415, 4,950, and 11,175 initial pairs, accounting for less than 1.0% of the total possible pairs. The AL process spans 18 iter-

| Datasets | Full Statistics | | Initial Seed Set ($t = 0$) | | Active Query Batch ($t > 0$) | |
|---|---|---|---|---|---|---|
| | # Graphs | # Pairs | # Graphs ($|\mathcal{G}_0|$) | # Pairs ($|\mathbb{L}_0|$) | # Graphs ($m$) | # Pairs ($|\Delta_t|$) |
| AIDS | 700 | 244,650 | 70 | 2,415 | 35 | 595 |
| LINUX | 1,000 | 499,500 | 100 | 4,950 | 50 | 1,225 |
| IMDB | 1,500 | 1,124,250 | 150 | 11,175 | 75 | 2,775 |

Table 1: Statistical summary and active learning configurations of used datasets.

ations. In each iteration $t$, after training the model to convergence on $\mathbb{L}_{t-1}$, we select $m$ informative graphs using our graph-centric strategy. To maintain consistency with the initialization phase, the incremental labeled set $\Delta_t$ is formed by densely pairing graphs within the selected batch $\mathcal{G}_t$. This yields $|\Delta_t| = \binom{m}{2}$ new labeled pairs per round. We set $m$ to 35, 50, and 75 for the three datasets, corresponding to 595, 1,225, and 2,775 new pairs per iteration. For a fair comparison with baselines that employ pair-level query strategies, we strictly control the budget by selecting exactly $|\Delta_t|$ pairs. This configuration demonstrates high data efficiency: even after 18 iterations, the cumulative labeled pairs remain below 5.5% of the total available pairs. Table 1 provides a comprehensive summary of these statistics.

**Metrics.** Following standard GSL protocols (Bai et al., 2019; Tan et al., 2023), we evaluate performance using three key metrics: Mean Squared Error (MSE) for regression accuracy, Spearman's Rank Correlation Coefficient ($\rho$) (Spearman, 1904) for ranking consistency, and Precision at 10 (P@10) for search quality.

**GSL Backbones.** To validate the model-agnostic nature of EVGS, we instantiate it with two representative backbones featuring distinct paradigms: the histogram-aggregation-based SimGNN (Bai et al., 2019) and the attention-aggregation-based NA-GSL (Tan et al., 2023). This diversity allows us to verify the framework's generalizability across different architectural strategies.

**Implementation Details.** We implement EVGS using PyTorch Geometric 2.4.0 on an NVIDIA RTX 3080 GPU. To ensure fair comparison, we strictly adhere to the original architectural settings for the SimGNN (Bai et al., 2019) and NA-GSL (Tan et al., 2023) backbones. All models are optimized via Adam (learning rate = 0.001, batch size = 128). Regarding EVGS-specific hyperparameters, we set the balancing coefficients $\lambda_1 = 0.01$ and $\lambda_2 = 0.1$ in equation 9 based on a grid search. The reference subset size is fixed at $K = 10$ to balance efficiency and estimation accuracy. Finally, we report the mean performance over 5 independent runs, with standard deviations visualized as shaded regions.

## 5.2 Performance Evaluation

### 5.2.1 Analysis on Query Strategies: Pairs vs. Graphs

To validate the superiority of our graph-centric variance-based selection, we conduct an ablation study using SimGNN on the AIDS dataset. We compare EVGS against two variants: **(1) EPS (Evidential Pair Selection)**, a pair-level strategy that directly selects pairs with the highest epistemic uncertainty $\mathcal{U}_{\text{epi}}$; and **(2) EMGS (Evidential Mean for Graph Selection)**, a graph-centric strategy that selects graphs maximizing the *mean* uncertainty of their associated pairs, rather than the variance.

As shown in Figures 1(a)-1(c), EPS consistently underperforms EVGS. Figure 1(d) reveals the underlying cause via t-SNE visualization: EPS suffers from a "hub effect." By myopically focusing on high-uncertainty pairs, it repeatedly samples pairs connected to the same "hub" graphs. Quantitatively, despite a budget of 35 graphs, EPS yields only **19 unique graphs**, resulting in redundant coverage and a failure to explore the broader data manifold. In contrast, EVGS explicitly optimizes for graph diversity, ensuring a uniform span of the similarity space. This demonstrates that treating pairs as independent atomic units is suboptimal; instead, graph-centric selection is essential to maximize the marginal information gain of each query.

These results also show that *variance* (EVGS) consistently outperforms the *mean* (EMGS). While EMGS prioritizes graphs exhibiting consistently high uncertainty across all pairings, our analysis suggests these

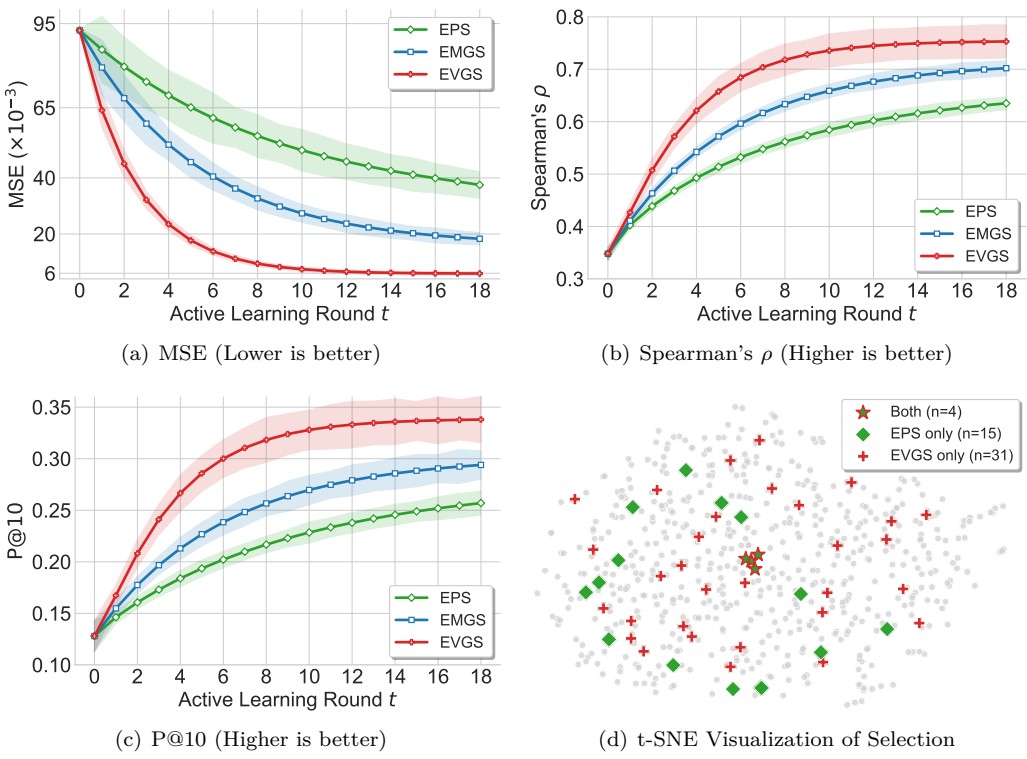

Figure 1: **Ablation study on query strategies** using SimGNN on the AIDS dataset.

often correspond to samples with inherently ambiguous substructures. The uncertainty associated with such samples is likely irreducible, implying that querying them yields diminishing returns. In contrast, EVGS targets graphs with high variance, identifying instances where the model is confident in some contexts but ambiguous in others. This high variance indicates that the graph lies in a transition region of the metric space, where the learned similarity function is most sensitive to perturbations. By labeling these pivotal instances, EVGS effectively refines the local geometry of the embedding manifold, thereby improving generalization.

### 5.2.2 Impact of Uncertainty Quantification Mechanisms

To validate the efficacy of our evidential regression framework and its components, we conduct an ablation study using NA-GSL on the LINUX dataset. We compare EVGS against two categories of variants:

**(1) External Uncertainty Baselines.** We replace our evidential framework with established uncertainty estimation methods while keeping the graph-centric selection strategy:

- **MVGS (MC-Dropout):** Estimates uncertainty via prediction variance over 25 stochastic forward passes (Gal & Ghahramani, 2016; Beluch et al., 2018).
- **DVGS (Deep Ensemble):** Estimates uncertainty using an ensemble of 3 independently trained models (Lakshminarayanan et al., 2017; Choi et al., 2021).

**(2) Internal Ablations.** We evaluate specific loss terms and uncertainty metrics within EVGS:

- **EVGS w/o MSE:** Removes loss term $\mathcal{L}_{\text{MSE}}$ to isolate the effect of the proposed regularization.
- **EVGS-A (Aleatoric):** Selects graphs based on aleatoric uncertainty ($\mathcal{U}_{\text{ale}}$).
- **EVGS-Total:** Uses total uncertainty ($\mathcal{U}_{\text{total}} = \mathcal{U}_{\text{ale}} + \mathcal{U}_{\text{epi}}$) as the selection metric.

As shown in Figures 2(a)-2(c), EVGS consistently outperforms all variants. First, the inferior performance of EVGS-A validates our premise: aleatoric uncertainty reflects inherent data noise rather than epistemic

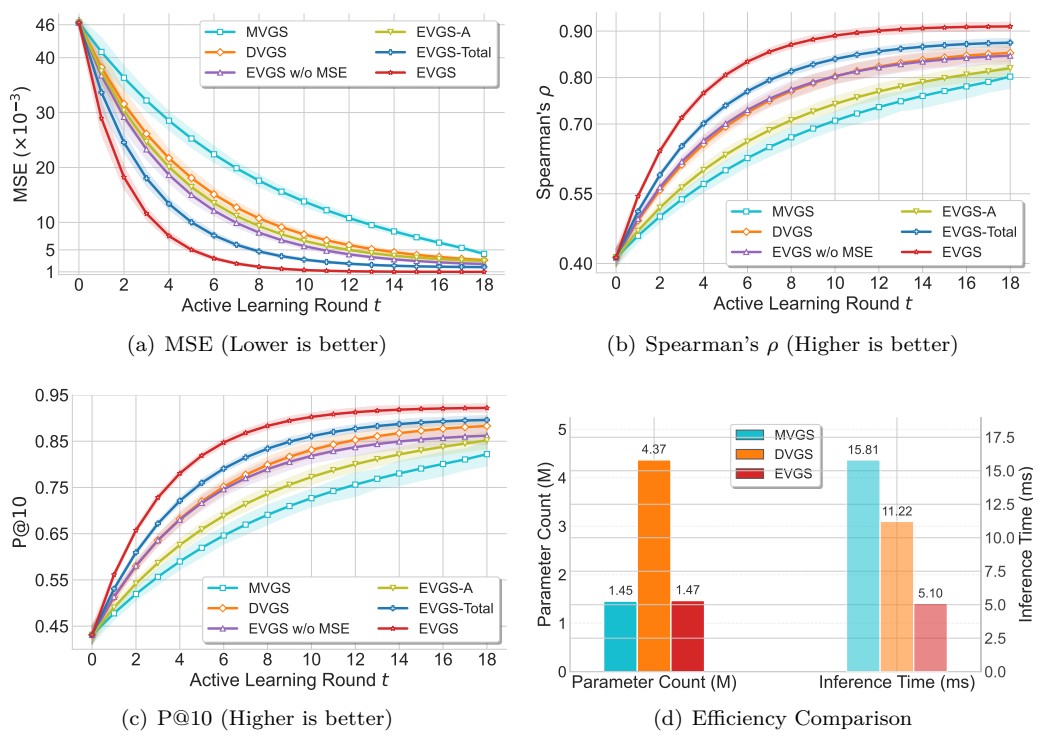

Figure 2: **Ablation study on uncertainty quantification** using NA-GSL on the LINUX dataset.

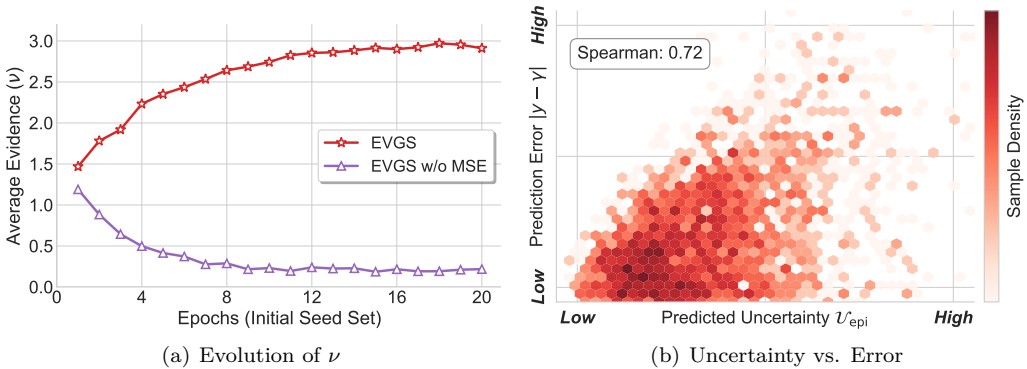

Figure 3: Empirical analysis of EVGS in **(a) training stability** and **(b) uncertainty estimation**.

ignorance. In an AL context, querying samples with high aleatoric uncertainty is inefficient, as the budget is squandered on irreducible noise rather than learnable structures. Consequently, EVGS-Total also degrades due to contamination from this noise component.

Second, the performance drop in EVGS w/o MSE underscores the necessity of the regularization term. To explicitly demonstrate its internal mechanism, we track the average predicted evidence ($\nu$) during the initial training phase on the seed set (Figure 3(a)). As observed, without the MSE anchor, the evidential model quickly falls into a "low-evidence trap" where $\nu$ collapses to near zero. As theoretically derived in Section 4.1.2, the gradient of the NLL loss with respect to the predictive mean $\gamma$ is directly proportional to the evidence (i.e., $\frac{\partial \mathcal{L}_{\text{NLL}}}{\partial \gamma} \propto \nu$). Consequently, this collapse of $\nu$ mathematically forces these specific gradients to vanish, halting the learning process of the target variable right from the start. Such an early failure is fatal, as it permanently derails the model's optimization and severely degrades all subsequent AL cycles. In contrast, including the MSE anchor effectively maintains a healthy level of evidence, preventing

vanishing gradients and ensuring stable optimization. Furthermore, to verify that the well-trained EVGS model produces reliable uncertainty estimates, we visualize the relationship between the predicted epistemic uncertainty ($\mathcal{U}_{\text{epi}}$) and the absolute prediction error ($|y - \gamma|$) in Figure 3(b). The clear positive correlation visually confirms that $\mathcal{U}_{\text{epi}}$ accurately reflects the model's true ignorance. In our graph-centric active learning framework, this reliability is paramount: it ensures that the uncertainty-derived scores (i.e., the variance of uncertainty) faithfully represent the actual error distribution across the graph, firmly justifying $\mathcal{U}_{\text{epi}}$ as a robust foundation for our acquisition metric.

Compared to external baselines, EVGS demonstrates superior efficacy and performance by addressing fundamental theoretical and computational limitations. Theoretically, MVGS and DVGS typically conflate aleatoric and epistemic uncertainties into a single predictive variance, lacking the granularity to filter out noisy samples. In contrast, EVGS leverages a higher-order conjugate prior to explicitly disentangle epistemic uncertainty within a single forward pass. Computationally, as shown in Figure 2(d), EVGS avoids the substantial parameter overhead of ensembles (DVGS) and the inference latency of stochastic sampling (MVGS). This establishes EVGS as a uniquely efficient solution for active graph similarity learning, delivering precise uncertainty estimation without incurring prohibitive resource costs.

### 5.2.3 Comparison with Baselines

**Baselines.** Given the scarcity of dedicated AL methods for graph similarity, we adapt representative strategies to this setting. Where applicable, we evaluate two variants: *Pair-level (P)*, operating on the joint pair embedding $\mathbf{z}$, and *Graph-centric (G)*, operating on individual graph embeddings $\mathbf{g}$.

- **Random:** Uniformly selects samples from the unlabeled pool. We report a unified baseline, as empirical results showed negligible differences between random-pair and random-graph selection.
- **Coreset (Sener & Savarese, 2018):** A diversity-based approach using $k$-Center Greedy ($k = 4$) to minimize the covering radius of the labeled set. We implement both **Coreset-P** and **Coreset-G**.
- **Cluster (Nguyen & Smeulders, 2004):** A density-based strategy using $k$-means ($k = 4$) to select representative instances near centroids. We evaluate both **Cluster-P** and **Cluster-G**.
- **BADGE (Ash et al., 2020):** A hybrid strategy that clusters samples in the gradient space to capture uncertainty and diversity. Inherently pair-centric, as gradients derive from the pairwise loss.

Figure 4 reports the AL performance on the IMDB dataset (additional datasets are detailed in Appendix B). Our proposed EVGS consistently outperforms all baselines across all metrics and backbones. Notably, EVGS exhibits the steepest learning curve in the initial rounds (0-6). For example, as shown in Figure 4(a), it reduces MSE significantly faster than BADGE and Coreset-G, demonstrating that evidential variance effectively identifies informative graphs for rapid adaptation. Furthermore, the narrow confidence intervals of EVGS indicate greater stability than the high variance observed in Random and Cluster-based methods.

## 5.3 In-depth Analysis of EVGS

### 5.3.1 Scalability of EVGS

To evaluate the scalability of EVGS in real-world large-scale scenarios, we conduct an in-depth analysis on the IMDB dataset. We design a "Controlled Pool Expansion" experiment to investigate how different selection strategies scale with an increasing number of candidate graphs. Specifically, during the acquisition phase, we artificially restrict the available unlabeled pool to proportions $r \in \{20\%, 40\%, 60\%, 80\%, 100\%\}$. We compare EVGS with two strong uncertainty-based baselines: MVGS (MC-Dropout) and DVGS (Deep Ensembles). The evaluation focuses on two critical dimensions: predictive performance (measured by the final MSE) and computational efficiency (measured by the acquisition time per round).

First, regarding *algorithmic scalability* (Figure 5(a)), we observe that as the candidate pool ratio $r$ increases, the MSE of all methods decreases, as a larger pool provides a broader search space for informative samples. However, EVGS consistently maintains the lowest MSE across all pool sizes. This demonstrates that our acquisition metric effectively leverages the expanded search space to pinpoint highly valuable graphs, rather than being distracted by the growing number of redundant or noisy candidates in a massive pool.

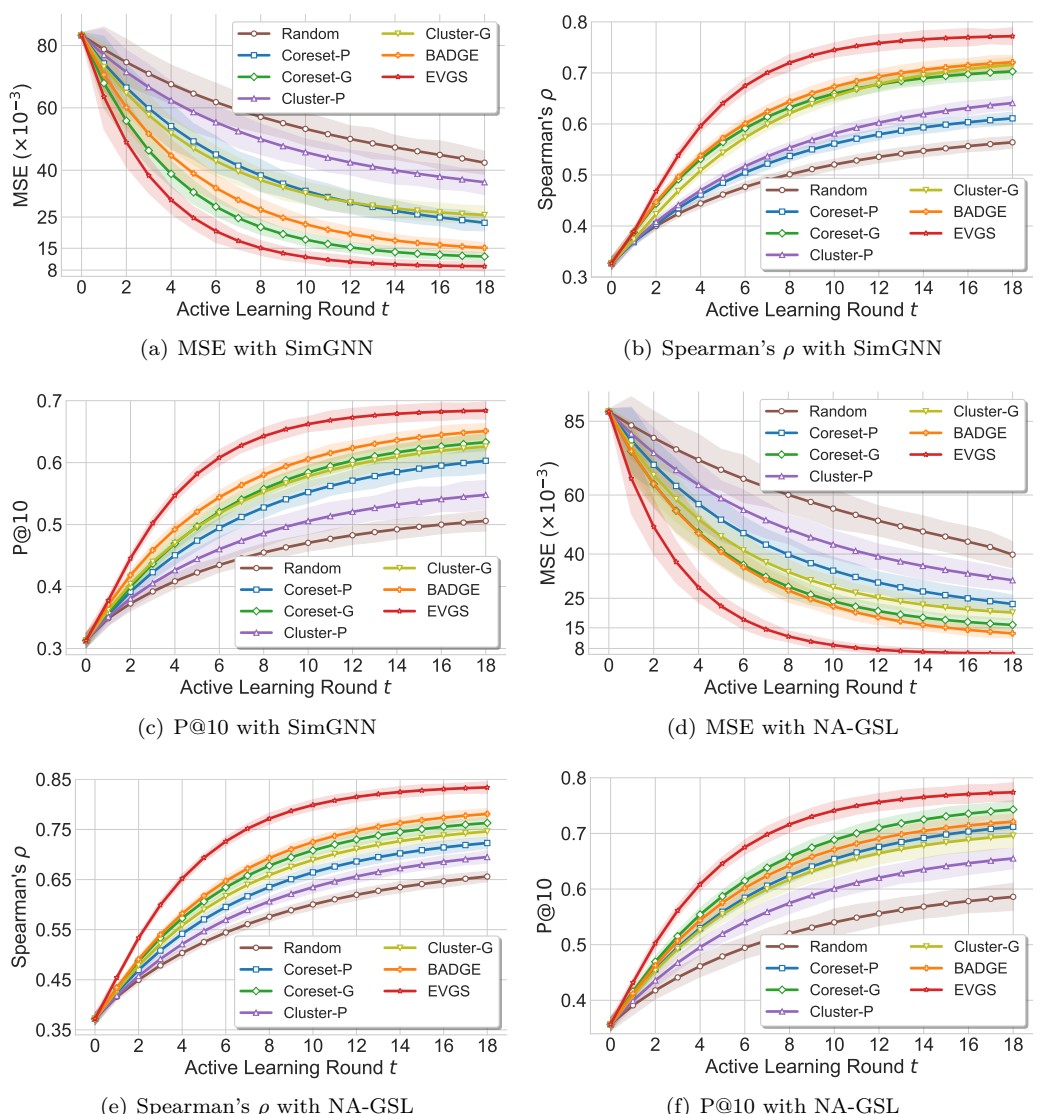

(a) MSE with SimGNN

(b) Spearman's $\rho$ with SimGNN

(c) P@10 with SimGNN

(d) MSE with NA-GSL

(e) Spearman's $\rho$ with NA-GSL

(f) P@10 with NA-GSL

Figure 4: **Performance comparison with baselines** on the IMDB dataset.

Second, regarding *computational scalability* (Figure 5(b)), EVGS exhibits an overwhelming advantage in time efficiency. Naturally, as the candidate pool size grows, the acquisition time for all methods increases. However, traditional uncertainty estimation methods such as MVGS and DVGS require multiple forward passes per graph pair, leading to remarkably steep slopes. In stark contrast, EVGS derives predictive uncertainty analytically via a *single forward pass* within the Evidential Regression framework. Consequently, its time cost scales with an extremely flat slope. This "fan-shaped" divergence in acquisition time confirms that EVGS is highly scalable and exceptionally well-suited for massive graph datasets.

### 5.3.2 Analysis of Selected Graphs

To provide qualitative insight into the efficacy of our graph-centric strategy, Figure 6 visualizes representative graphs selected by EVGS. Driven by our evidential variance metric, the selected subset exhibits significant topological diversity. This graph-centric diversity naturally yields training pairs spanning a broad spectrum of ground-truth similarities. For example, the selected pool contains graphs that form highly structurally

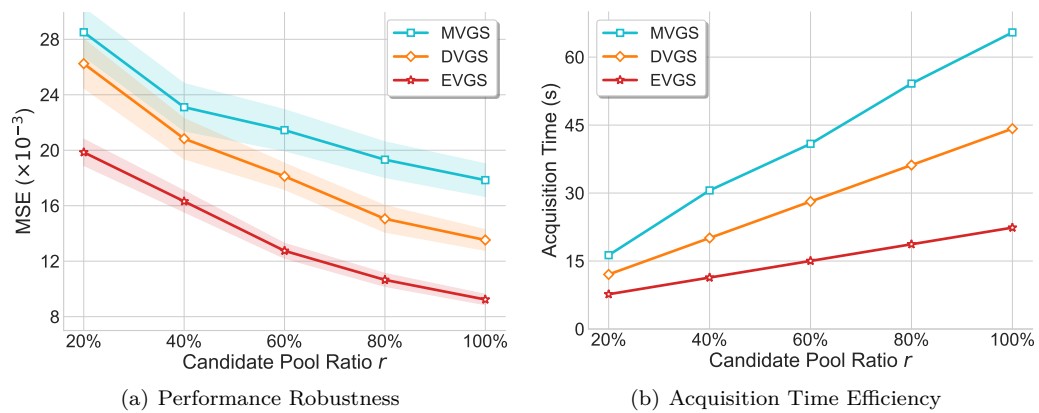

(a) Performance Robustness

(b) Acquisition Time Efficiency

Figure 5: **Scalability analysis of EVGS**: (a) performance and (b) efficiency.

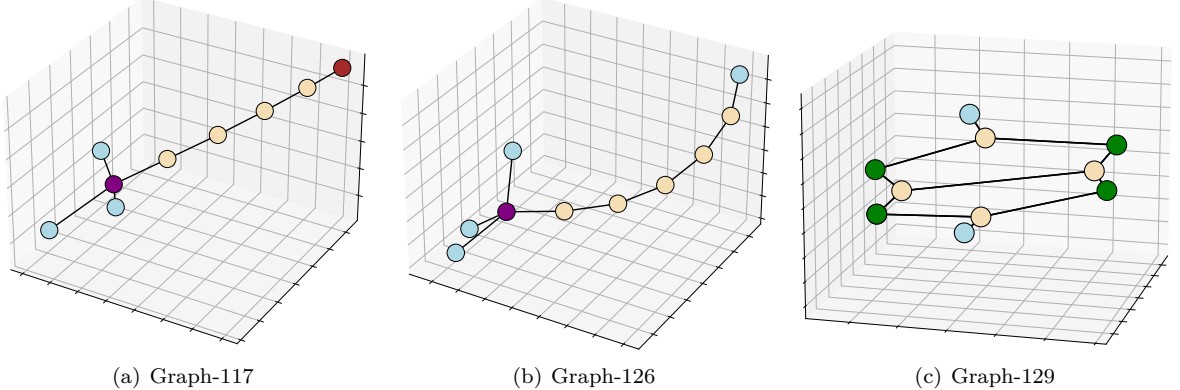

(a) Graph-117

(b) Graph-126

(c) Graph-129

Figure 6: **Visualization of representative graphs selected by EVGS.**

isomorphic pairs (e.g., Figure 6(a) and Figure 6(b)) as well as structurally distinct pairs (e.g., Figure 6(b) and Figure 6(c)). This diversity is instrumental for efficient model training in two critical aspects:

- **Metric Calibration via Anchors:** By querying pairs at both ends of the similarity spectrum (i.e., highly similar vs. highly dissimilar), EVGS provides the model with critical "anchors" to calibrate the regression scale. This effectively prevents the model from collapsing into trivial solutions or exhibiting bias towards the mean similarity score.

- **Structural Manifold Coverage:** The distinct topological variations observed in the selected graphs confirm that EVGS effectively explores diverse regions of the graph space. Unlike baselines that suffer from the "hub effect" (i.e., repeatedly selecting a redundant set of certain graphs), our method identifies informative instances that broadly cover the underlying data manifold, thereby accelerating the generalization capability of the GSL backbone.

To empirically validate our analysis regarding the pathological behavior of pair-level selection strategies, we investigate the topological properties of the graph subsets queried by different methods. Specifically, we compare our EVGS against the pair-level strategy (EPS) and the overall data pool. We employ graph size (number of nodes) and average degree as two fundamental proxies for structural complexity.

As illustrated in Figure 7, a stark contrast emerges between the two AL strategies. The subset selected by EPS deviates significantly from the true distribution, heavily concentrating on the upper bounds. This visual evidence perfectly captures the "hubness trap": EPS is easily misled by dense, ambiguous substructures, wasting the query budget on highly redundant, complex graphs. Conversely, our proposed EVGS exhibits a distinct, uniform distribution that spans the entire range of structural metrics, from extremely sparse graphs

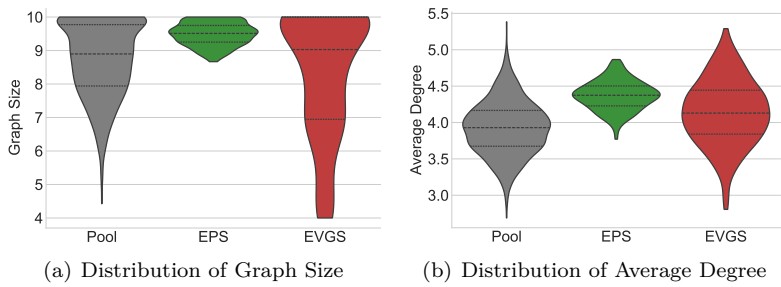

(a) Distribution of Graph Size  (b) Distribution of Average Degree

Figure 7: Distributions of **(a) graph size** and **(b) average degree** for the selected graph subsets.

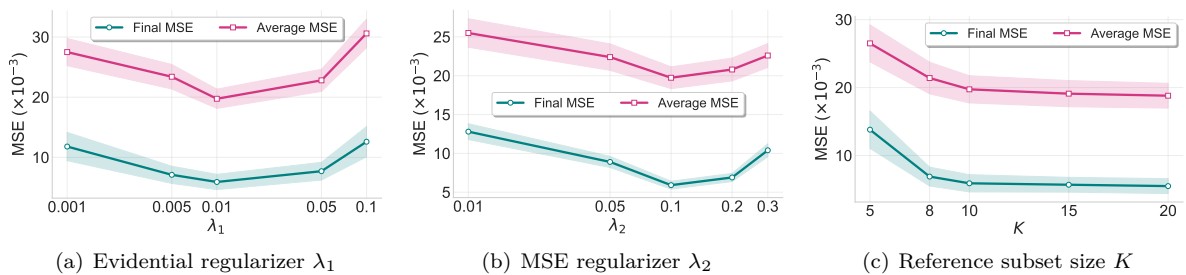

(a) Evidential regularizer $\lambda_1$  (b) MSE regularizer $\lambda_2$  (c) Reference subset size $K$

Figure 8: **Sensitivity analysis of EVGS** using SimGNN on the AIDS dataset.

to dense substructures. Rather than collapsing into local topological traps, EVGS successfully selects a diverse set of graphs that act as *topological anchors*. These phenomena strongly corroborate our motivation: unlike pair-level strategies that are vulnerable to complex topologies, EVGS uniformly covers the structural manifold, thereby providing an unbiased and comprehensive structural calibration for the model.

### 5.3.3 Hyperparameter Sensitivity

To investigate the stability of EVGS, we conduct a sensitivity analysis on three key hyperparameters: the evidential regularizer coefficient $\lambda_1$, the MSE regularizer coefficient $\lambda_2$, and the reference subset size $K$. We report both the *Final MSE* (at the last active learning round) and the *Average MSE* (across all rounds) on the AIDS dataset using the SimGNN backbone. The results are illustrated in Figure 8.

**Impact of Evidential Regularization Coefficient $\lambda_1$.** Figure 8(a) displays the performance variation concerning the evidential regularizer $\lambda_1$. We observe a convex pattern: small values ($\lambda_1 < 0.005$) lead to under-regularization, where the model fails to penalize misleading evidence, while excessive regularization ($\lambda_1 > 0.05$) forces the distribution towards a uniform prior, hindering representation learning. The performance is robust within $[0.005, 0.05]$, peaking at $\lambda_1 = 0.01$.

**Impact of MSE Regularizer Coefficient $\lambda_2$.** Figure 8(b) examines the sensitivity to $\lambda_2$, which serves as the critical gradient anchor coefficient. For small values $\lambda_2$ (e.g., $< 0.05$), the model lacks sufficient gradient injection to counteract the vanishing gradient pathology inherent to evidential regression. As detailed in our theoretical derivation, without this anchor, the optimization tends to collapse into a "uniform ignorance" state ($\nu \to 0^+$) where NLL gradients vanish. This results in a flattened uncertainty landscape, effectively degrading the AL strategy to random sampling. Conversely, when $\lambda_2$ is excessively large (e.g., $> 0.2$), the deterministic MSE loss overpowers the probabilistic NLL objective. While this enforces regression consistency, it suppresses the fine-grained learning of evidential parameters, rendering the derived epistemic uncertainty uncalibrated for sample ranking. The performance optimality at $\lambda_2 = 0.1$ confirms that this setting provides the necessary gradient flow to prevent evidence collapse while preserving the dominance of the evidential objective for uncertainty quantification.

**Impact of Reference Subset Size $K$.** Figure 8(c) investigates the size of the reference subset $K$ used for graph-centric selection. While a larger $K$ theoretically provides a more robust estimation of uncertainty variance, it incurs higher computational overhead. We observe that the performance gain saturates around $K = 10$. Consequently, we fix $K = 10$ for all experiments to achieve an optimal trade-off between estimation reliability and computational efficiency.

## 6 Discussion

Although EVGS builds upon established concepts like evidential learning and uncertainty sampling, its true value lies in how it adapts and synergizes these elements to address the unique bottlenecks of AL in GSL. We highlight our core innovations across three perspectives:

**Task Perspective (Bridging AL and GSL).** Unlike traditional graph AL, which mainly focuses on discrete node classification, applying AL to GSL introduces two specific challenges: (1) *Uncertainty Quantification in Regression* and (2) *Structural Dependency in Paired Inputs*. EVGS provides a tailored framework to address these issues. By formalizing this new paradigm and addressing these challenges, EVGS provides a novel framework that benefits both the AL and graph representation learning communities.

**Methodological Perspective (Adapting Evidential Regression).** To address the first challenge efficiently, we adapt evidential regression for uncertainty estimation. Observing that standard evidential models can suffer from uncertainty degradation in low-data AL regimes, we introduce a targeted regularization strategy to ensure more robust uncertainty quantification for AL.

**Algorithmic Perspective (Graph-centric Sampling).** To address the second challenge, we note that independently selecting high-uncertainty pairs often leads to structural redundancy (the "Hubness" trap). Consequently, EVGS employs a *graph-centric* acquisition strategy based on uncertainty variance, explicitly modeling structural interactions among candidate pairs to improve sampling efficiency.

### Limitation

Regarding scalability, while our graph-centric acquisition approximation successfully improves efficiency with respect to the number of candidate graphs ($N$), reducing the combinatorial complexity from $O(N^2)$ to $O(NK)$, the computational cost remains dependent on graph size. Specifically, scalability for large individual graphs is inherently tied to the chosen GSL backbone. Although EVGS is backbone-agnostic and could employ lightweight GSL models to handle massive graphs, its scalability and effectiveness have not yet been empirically validated on very large individual graphs.

## 7 Conclusion

This work pioneers the integration of AL with GSL, establishing a data-efficient paradigm to address the prohibitive annotation costs in GSL. We introduce EVGS, a generalizable framework that overcomes two fundamental challenges: it quantifies regression uncertainty via evidential deep learning with MSE regularization, and resolves the combinatorial complexity of paired inputs through a novel graph-centric selection strategy based on uncertainty variance. Extensive experiments across three benchmarks and two GSL architectures demonstrate that EVGS consistently outperforms established baselines. By shifting the focus from pairwise to graph-centric informativeness, our work offers fresh insights into the structural redundancy in GSL datasets and paves a new avenue for cost-effective graph representation learning.

## Acknowledgment

This work was supported by the National Natural Science Foundation of China (Grant No. 62572097).

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

# A    Detailed Theoretical Derivation

We provide the detailed mathematical derivations for the uncertainty decomposition and the marginal likelihood objective used in our evidential framework. Let the predicted evidential parameters be $\mathbf{o} = \{\gamma, \nu, \alpha, \beta\}$. The Normal-Inverse-Gamma (NIG) prior over the unknown mean $\mu$ and variance $\sigma^2$ is defined as:

$$p(\mu, \sigma^2 \mid \mathbf{o}) = \mathcal{N}\left(\mu \mid \gamma, \frac{\sigma^2}{\nu}\right) \Gamma^{-1}(\sigma^2 \mid \alpha, \beta). \tag{11}$$

## A.1    Derivation of Epistemic Uncertainty

The epistemic uncertainty $\mathcal{U}_{\text{epi}}$ captures the model's lack of knowledge about the true mean $\mu$, which is quantified by the variance of $\mu$. We derive this rigorously using the *Law of Total Variance*:

$$\mathcal{U}_{\text{epi}} = \text{Var}[\mu] = \mathbb{E}_{\sigma^2}[\text{Var}[\mu \mid \sigma^2]] + \text{Var}_{\sigma^2}[\mathbb{E}[\mu \mid \sigma^2]]. \tag{12}$$

From the conditional Gaussian distribution $(\mu \mid \sigma^2) \sim \mathcal{N}(\gamma, \sigma^2/\nu)$, we can determine the inner terms:

1. $\mathbb{E}[\mu \mid \sigma^2] = \gamma$. Since $\gamma$ is a deterministic parameter output by the neural network and is constant with respect to $\sigma^2$, its variance is zero: $\text{Var}_{\sigma^2}[\gamma] = 0$.

2. $\text{Var}[\mu \mid \sigma^2] = \frac{\sigma^2}{\nu}$.

Substituting these into the total variance equation, the second term vanishes, yielding:

$$\text{Var}[\mu] = \mathbb{E}_{\sigma^2}\left[\frac{\sigma^2}{\nu}\right] = \frac{1}{\nu}\mathbb{E}_{\sigma^2}[\sigma^2]. \tag{13}$$

To compute $\mathbb{E}_{\sigma^2}[\sigma^2]$, we use the probability density function of the Inverse-Gamma distribution $\Gamma^{-1}(\alpha, \beta)$:

$$\mathbb{E}_{\sigma^2}[\sigma^2] = \int_0^\infty \sigma^2 \frac{\beta^\alpha}{\Gamma(\alpha)} (\sigma^2)^{-(\alpha+1)} \exp\left(-\frac{\beta}{\sigma^2}\right) d\sigma^2 = \frac{\beta^\alpha}{\Gamma(\alpha)} \int_0^\infty (\sigma^2)^{-\alpha} \exp\left(-\frac{\beta}{\sigma^2}\right) d\sigma^2. \tag{14}$$

Recognizing the integral part as the unnormalized form of an Inverse-Gamma distribution with parameters $(\alpha - 1, \beta)$, it evaluates to $\frac{\Gamma(\alpha-1)}{\beta^{\alpha-1}}$. Thus, for $\alpha > 1$:

$$\mathbb{E}_{\sigma^2}[\sigma^2] = \frac{\beta^\alpha}{\Gamma(\alpha)} \frac{\Gamma(\alpha-1)}{\beta^{\alpha-1}} = \frac{\beta}{\alpha - 1}. \tag{15}$$

Substituting this back, the epistemic uncertainty is analytically derived as:

$$\mathcal{U}_{\text{epi}} = \frac{\beta}{\nu(\alpha - 1)}. \tag{16}$$

## A.2    Derivation of the Marginal Likelihood and Learning Objective

To formulate the learning objective, we compute the marginal likelihood $p(y \mid \mathbf{o})$ by integrating out the latent parameters $\mu$ and $\sigma^2$:

$$p(y \mid \mathbf{o}) = \int_0^\infty \int_{-\infty}^\infty p(y \mid \mu, \sigma^2) p(\mu, \sigma^2 \mid \mathbf{o}) \, d\mu \, d\sigma^2. \tag{17}$$

**Step 1: Marginalizing out $\mu$.** Since the likelihood is $(y \mid \mu, \sigma^2) \sim \mathcal{N}(\mu, \sigma^2)$ and the prior is $(\mu \mid \sigma^2) \sim \mathcal{N}(\gamma, \sigma^2/\nu)$, the convolution of these two Gaussian distributions results in another Gaussian distribution for $y$ conditioned only on $\sigma^2$:

$$p(y \mid \sigma^2, \mathbf{o}) = \int_{-\infty}^\infty \mathcal{N}(y \mid \mu, \sigma^2) \mathcal{N}\left(\mu \mid \gamma, \frac{\sigma^2}{\nu}\right) d\mu = \mathcal{N}\left(y \mid \gamma, \sigma^2\left(1 + \frac{1}{\nu}\right)\right). \tag{18}$$

**Step 2: Marginalizing out $\sigma^2$.** We integrate the product of this new Gaussian and the Inverse-Gamma prior. Expanding the probability density functions, we get:

$$p(y \mid \mathbf{o}) = \int_0^\infty \frac{1}{\sqrt{2\pi\sigma^2(1+1/\nu)}} \exp\left(-\frac{(y-\gamma)^2}{2\sigma^2(1+1/\nu)}\right) \frac{\beta^\alpha}{\Gamma(\alpha)}(\sigma^2)^{-(\alpha+1)} \exp\left(-\frac{\beta}{\sigma^2}\right) d\sigma^2. \tag{19}$$

By grouping the terms involving $\sigma^2$, we can rewrite the integrand:

$$p(y \mid \mathbf{o}) = \frac{\beta^\alpha}{\Gamma(\alpha)\sqrt{2\pi(1+1/\nu)}} \int_0^\infty (\sigma^2)^{-\left(\alpha+\frac{1}{2}+1\right)} \exp\left(-\frac{1}{\sigma^2}\left(\beta + \frac{\nu(y-\gamma)^2}{2(1+\nu)}\right)\right) d\sigma^2. \tag{20}$$

The integral is exactly the unnormalized form of an Inverse-Gamma distribution with updated parameters $\tilde{\alpha} = \alpha + 1/2$ and $\tilde{\beta} = \beta + \frac{\nu(y-\gamma)^2}{2(1+\nu)}$. Evaluating this integral yields $\Gamma(\tilde{\alpha})/\tilde{\beta}^{\tilde{\alpha}}$. Therefore:

$$p(y \mid \mathbf{o}) = \frac{\Gamma(\alpha+1/2)}{\Gamma(\alpha)\sqrt{2\pi(1+1/\nu)}} \frac{\beta^\alpha}{\left(\beta + \frac{\nu(y-\gamma)^2}{2(1+\nu)}\right)^{\alpha+1/2}}. \tag{21}$$

**Connection to the Learning Objective.** By factoring out $\beta$ in the denominator, the expression matches the standard probability density function of a Student-t distribution $\text{St}(y; \tilde{\mu}, \tilde{\sigma}^2, \tilde{\nu})$:

$$p(y \mid \mathbf{o}) = \text{St}\left(y; \gamma, \frac{\beta(1+\nu)}{\nu\alpha}, 2\alpha\right), \tag{22}$$

where the location parameter is $\gamma$, the scale parameter is $\frac{\beta(1+\nu)}{\nu\alpha}$, and the degrees of freedom is $2\alpha$.

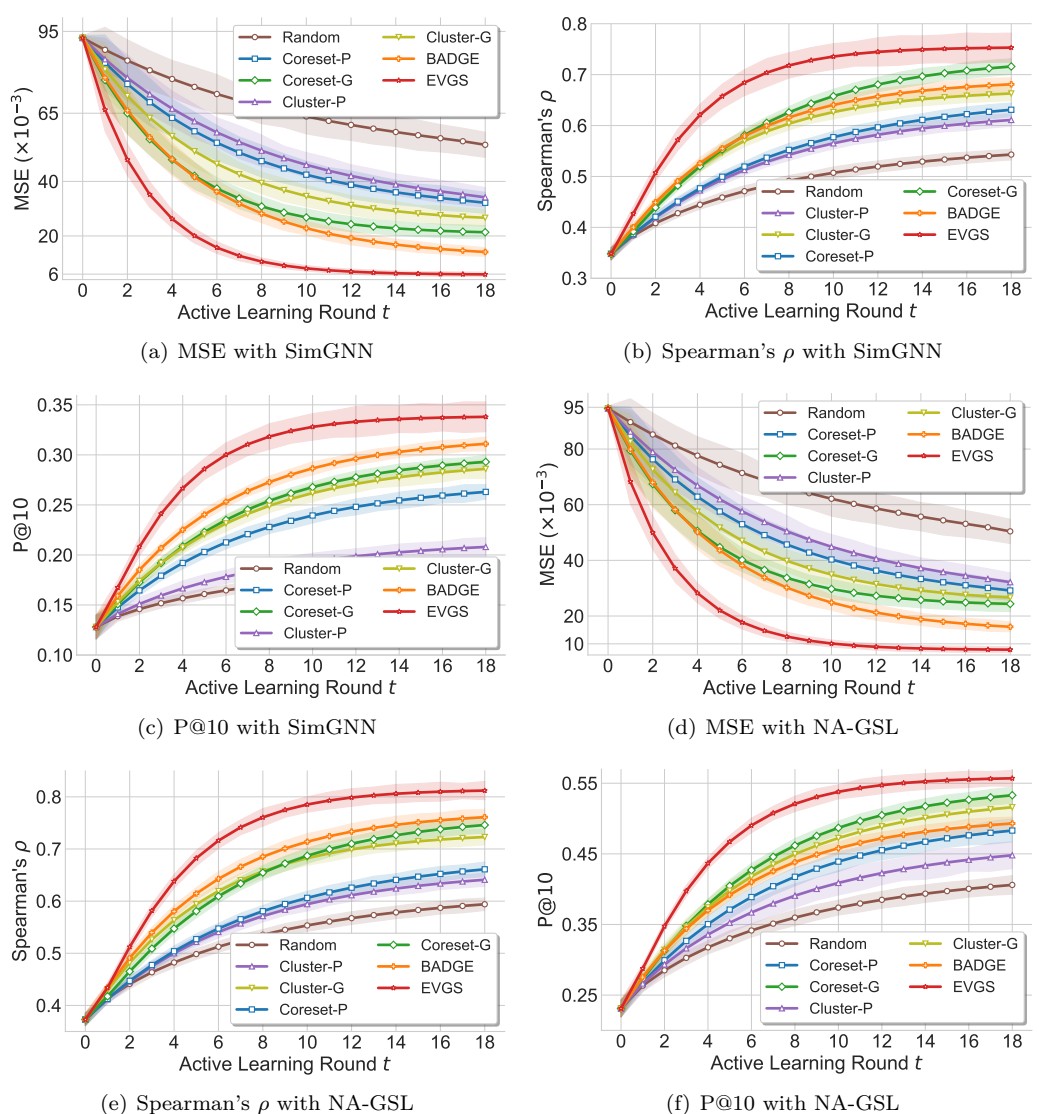

Figure 9: **Performance comparison with baselines** on the AIDS dataset.

## B Comprehensive Baseline Comparisons

Performance comparisons on the AIDS and LINUX datasets are presented in Figure 9 and Figure 10, respectively. These results corroborate the trends reported in the main paper, demonstrating the generalizability of our EVGS across diverse graph domains.

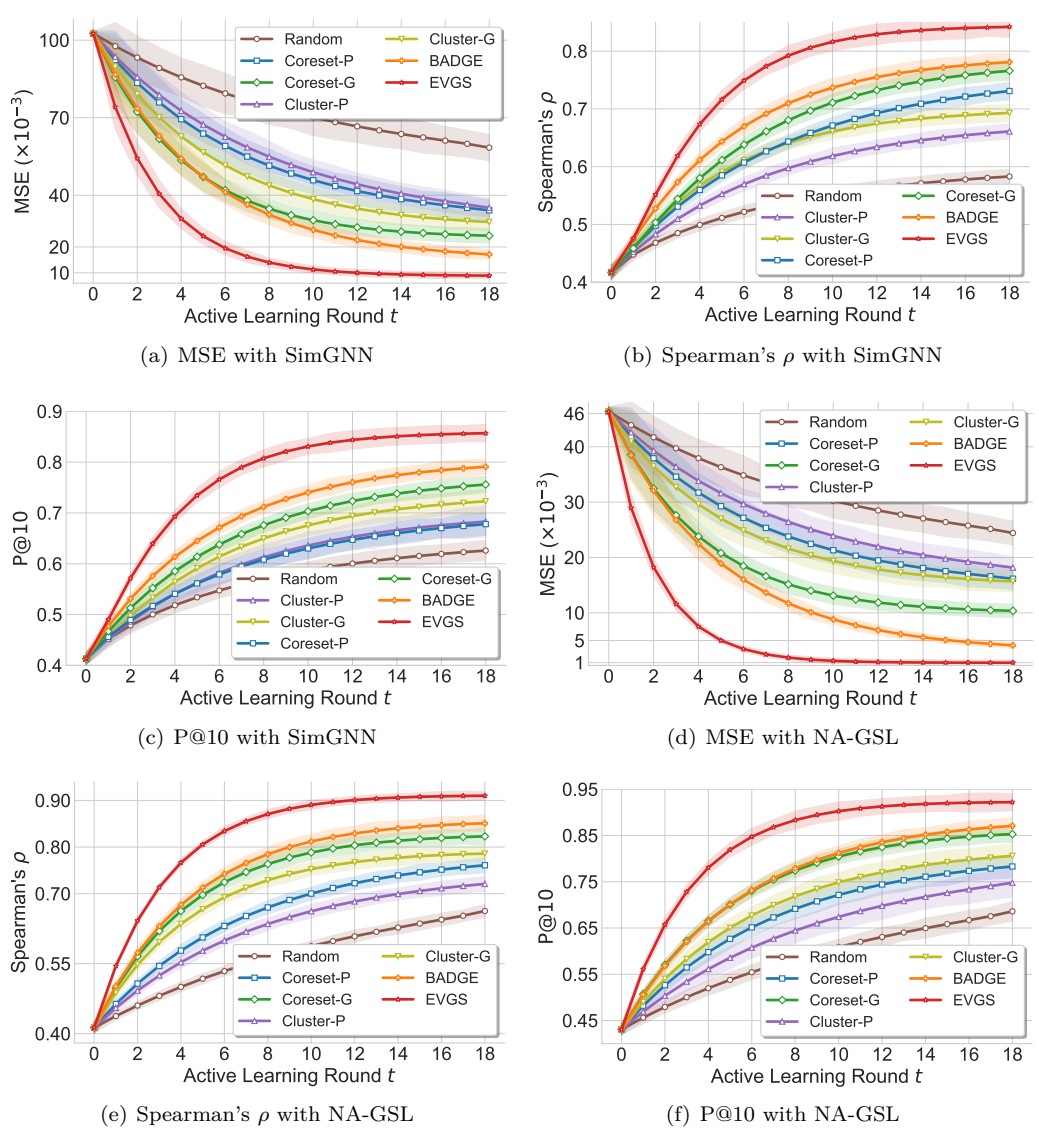

Figure 10: **Performance comparison with baselines** on the LINUX dataset.

