# OpenReview forum: "When Active Learning Meets Graph Similarity: Evidential Variance for Graph Selection"
_TMLR — Accepted by TMLR_

### Review · Reviewer_XoxF · 2026-03-03

**Summary Of Contributions:**

This paper proposes to adapt active learning method (AL) for graph similarity learning (GSL). From a high level perspective, to solve the expensive data labeling, the AL method first labels a small subset of labels and cumulatively label more data that is useful for model performance. The proposed EVGS method serves as a rigorous theory background for the graph centric sampling method in the pool-based batch-mode active learning paradigm. The experiment results also show the consistent improvment of the proposed method EVGS over the baselines.

**Audience:**

Yes

**Audience Explanation:**

AL for GSL application seems a novel topic, and both graph learning and AL community will benefit from it.

**Claims And Evidence:**

Yes

**Claims Explanation:**

1. The labeling expense is drastically lower with active learning, by initially labeling a small subset of graphs, and executing a couple of AL rounds, which solves the initial problem of GSL labeling.
2. The experiments on query strategies shows the significance of EVGS over pair-wise method EPS and another graph centric candidate EMGS.
3. EVGS is consistently better compared against other uncertainty methods MVGS and DVGS.
4. EVGS is stronger than other AL baselines, both pair and graph centric methods.

**Requested Changes:**

The crucial part is the derivation of probabilistic theory in section 4, which is a bit over-simplified. For instance, from the definition, graphs are from a Gaussian distribution, but I don't know how exactly does a Gaussian distribution sample a graph. By introducting NIG, what is the exact derivation for distribution simplification, and how to calculate the epistemic uncertainty? The text description and the mathematic formulars are just introductory and lack of depth. It might be common knowledge for AL community, but it would be better to elaborate this for the completeness of the paper, and for readers from other fields.

---

> ### Author Response · Authors · 2026-03-26
>
> We sincerely thank the reviewer for the positive feedback, the recognition of our method's significance, and the constructive suggestions to enhance the mathematical rigor of our paper. We fully agree that a detailed derivation is crucial for completeness and for readers from diverse backgrounds.
>
> In the revised manuscript, we have expanded our mathematical explanations. All corresponding modifications have been highlighted in **Brick Red** in the updated PDF. Below, we address your specific questions:
>
> **1. Clarification on the Gaussian Assumption**
>
> *   **Reviewer's Concern:** "...graphs are from a Gaussian distribution, but I do not know how exactly a Gaussian distribution samples a graph."
> *   **Response:** We apologize for the ambiguity in our original text. We have explicitly clarified in Section 4.1.1 that the Gaussian distribution does not model the generative process of the discrete graphs themselves. Instead, it models the continuous scalar similarity score $y \in \mathbb{R}$ between a given graph pair $(G, G')$. Treating a continuous target variable as a sample drawn from a Gaussian distribution with an unknown mean and variance is a standard and well-established formulation in Deep Evidential Regression literature [1, 2, 3, 4]. We have updated the text to make this distinction clear.
>
> **2. Calculation of Epistemic Uncertainty**
>
> *   **Reviewer's Concern:** "...how to calculate the epistemic uncertainty?"
> *   **Response:** To address this, we have added a detailed, step-by-step derivation in Appendix A.1. We rigorously derive the epistemic uncertainty using the Law of Total Variance. We explicitly show how the inner terms $\mathbb{E}[\mu \mid \sigma^2]$ and $\operatorname{Var}[\mu \mid \sigma^2]$ are computed from the conditional Gaussian, and how integrating over the Inverse-Gamma prior $\Gamma^{-1}(\alpha, \beta)$ analytically yields the final uncertainty quantification $\mathcal{U}_{\text{epi}} = \frac{\beta}{\nu(\alpha - 1)}$.
>
> **3. Exact Derivation for Distribution Simplification**
>
> *   **Reviewer's Concern:** "...what is the exact derivation for distribution simplification... The text description and the mathematical formulas are just introductory and lack depth."
> *   **Response:** We completely agree that the previous version was over-simplified. In Appendix A.2, we now provide the exact, analytical derivation of the marginal likelihood $p(y \mid \mathbf{o})$. We break down the double integral into two explicit steps:
>     *   **Marginalizing out $\mu$:** Showing the convolution of the two Gaussian distributions.
>     *   **Marginalizing out $\sigma^2$:** Showing the integration of the resulting Gaussian with the Inverse-Gamma prior, and how grouping the terms forms an unnormalized Inverse-Gamma distribution.
>
> We believe these detailed derivations provide the exact depth and completeness requested. We thank you again for helping us improve the quality of our paper.
>
> **References:**
> [1] Alexander Amini, Wilko Schwarting, Ava Soleimany, and Daniela Rus. Deep evidential regression. In Advances in Neural Information Processing Systems, pp. 14927–14937, 2020.
> [2] Dongpin Oh and Bonggun Shin. Improving evidential deep learning via multi-task learning. In Proceedings of the AAAI Conference on Artificial Intelligence, pp. 7895–7903, 2022.
> [3] Yuefei Wu, Bin Shi, Bo Dong, Qinghua Zheng, and Hua Wei. The evidence contraction issue in deep evidential regression: Discussion and solution. In Proceedings of the AAAI Conference on Artificial Intelligence, pp. 21726–21734, 2024.
> [4] Kai Ye, Tiejin Chen, Hua Wei, and Liang Zhan. Uncertainty regularized evidential regression. In Proceedings of the AAAI Conference on Artificial Intelligence, pp. 16460–16468, 2024.

---

### Review · Reviewer_Msma · 2026-03-08

**Summary Of Contributions:**

This paper studies how to reduce labeling costs in Graph Similarity Learning (GSL), where models predict similarity scores between pairs of graphs but typically require many labeled examples. The authors propose EVGS (Evidential Variance for Graph Selection), an active learning approach designed for this setting. The method estimates epistemic uncertainty using evidential deep learning and adds an MSE-based regularization term to stabilize training when labeled data is scarce. Instead of selecting individual graph pairs for annotation, EVGS selects informative graphs by measuring the variance of their uncertainty across interactions with other graphs, aiming to identify graphs that lie in ambiguous regions of the similarity space. Experiments on the AIDS, LINUX, and IMDB datasets show that the method improves performance compared to several active learning baselines while using only a small fraction of the possible labeled pairs.

**Audience:**

Yes

**Audience Explanation:**

The paper studies active learning for graph similarity learning, which is relevant to researchers working on graph machine learning, active learning, and uncertainty estimation. The problem of reducing labeling costs for graph-based tasks is important in many applications, and the proposed approach could be of interest to researchers exploring data-efficient training for graph models.

**Broader Impact Concerns:**

There are no broader impact concerns from this work.

**Claims And Evidence:**

No

**Claims Explanation:**

The paper proposes interesting ideas, but several of its claims are not supported by sufficiently convincing evidence. Much of the evidence relies on ablation experiments showing that removing certain components degrades performance. While this indicates that the components are useful, it does not fully validate the mechanisms proposed in the paper or clarify what specific problems these components address.

The authors attribute the effectiveness of their approach to the evidential regression formulation and the graph-centric uncertainty-variance acquisition strategy. The paper includes ablations showing that removing the MSE anchor degrades performance and provides a sensitivity analysis for its coefficient. However, the explanation motivating this component, namely gradient shrinkage in low-evidence regimes, is not empirically validated.

A similar issue appears in the motivation for the variance-based acquisition strategy. Although it outperforms several alternatives, the paper does not provide further analysis explaining why this strategy works better or whether the selected graphs are actually more informative.

Overall, the paper presents interesting ideas and promising empirical results, but several claims appear stronger than what the current evidence supports. The authors should either provide additional empirical analysis to support the proposed mechanisms or reduce the claims.

**Requested Changes:**

* Show empirical evidence for the gradient shrinkage issue (e.g., track gradient magnitudes or evidence values during training) to support the motivation for the MSE anchor.
* Analyze the uncertainty estimates (for example, correlation between prediction error and predicted epistemic uncertainty) to check whether the uncertainty used for acquisition is meaningful.
* Evaluate the method on datasets with larger graph pools to understand how the selection strategy scales as the number of candidate graphs increases.
* Provide some analysis of the samples selected during active learning (e.g., what kinds of graphs are chosen by the variance-based strategy) to better explain why the method improves performance.

---

> ### Author Response · Authors · 2026-03-26
>
> We sincerely thank you for your thoughtful review and for recognizing the relevance of our work to the graph machine learning and active learning communities, as well as the promising nature of our empirical results. We completely agree with your assessment that our proposed mechanisms require deeper empirical validation to fully substantiate our claims.
>
> Guided by your highly constructive feedback, we have conducted comprehensive additional analyses and experiments. We believe these additions directly address your concerns and significantly strengthen the paper. All corresponding modifications in the revised manuscript have been highlighted in **Teal**.
>
> Below is a detailed point-by-point response to your requested changes:
>
> ### 1. Empirical Evidence for the Gradient Shrinkage Issue
> **Reviewer's Request:** *Show empirical evidence for the gradient shrinkage issue (e.g., track gradient magnitudes or evidence values during training) to support the motivation for the MSE anchor.*
>
> **Our Response:** Thank you for this excellent suggestion. To explicitly demonstrate the internal mechanism and validate the gradient shrinkage pathology, we have added a new empirical analysis in **Section 5.2.2** and **Figure 3(a)**.
> Specifically, we tracked the evolution of the average predicted evidence ($\nu$) during the initial training phase. As theoretically derived in our methodology, the gradient of the NLL loss with respect to the predictive mean is directly proportional to $\nu$. As shown in Figure 3(a), without the MSE anchor, the evidential model quickly falls into a "low-evidence trap" where $\nu$ collapses to near zero, mathematically forcing the gradients to vanish and halting the learning process. In contrast, the inclusion of our MSE anchor effectively maintains a healthy level of evidence, preventing gradient shrinkage and ensuring stable optimization.
>
> ### 2. Analysis of the Uncertainty Estimates
> **Reviewer's Request:** *Analyze the uncertainty estimates (for example, correlation between prediction error and predicted epistemic uncertainty) to check whether the uncertainty used for acquisition is meaningful.*
>
> **Our Response:** We fully agree that validating the quality of the estimated uncertainty is crucial for an active learning framework. To address this, we have added a correlation analysis in **Section 5.2.2** and **Figure 3(b)**.
> We visualized the relationship between the predicted epistemic uncertainty and the absolute prediction error. The results (Figure 3(b)) demonstrate a clear, strong positive correlation (Spearman's correlation: 0.72). This confirms that our regularized evidential framework produces highly reliable uncertainty estimates that faithfully reflect the model's true ignorance, thereby firmly justifying the use of $\mathcal{U}_{epi}$ as the foundation for our acquisition metric.

---

> ### Author Response · Authors · 2026-03-26
>
> ### 3. Evaluation on Datasets with Larger Graph Pools
> **Reviewer's Request:** *Evaluate the method on datasets with larger graph pools to understand how the selection strategy scales with increasing candidate graph counts.*
>
> **Our Response:** To thoroughly investigate scalability, we have designed a new "Controlled Pool Expansion" experiment, detailed in the new **Section 5.3.1** and **Figure 5**.
> Using the IMDB dataset, we artificially varied the ratio of the available unlabeled candidate pool from 20\% to 100\%. We evaluated both algorithmic performance and computational efficiency:
> *   **Performance Robustness (Fig 5a):** As the pool size increases, EVGS consistently leverages the expanded search space to pinpoint valuable graphs, maintaining the lowest MSE across all pool sizes without being distracted by redundant candidates.
> *   **Time Efficiency (Fig 5b):** Compared to traditional baselines like MC-Dropout (MVGS) and Deep Ensembles (DVGS), whose acquisition times grow steeply, EVGS derives uncertainty analytically via a single forward pass. Consequently, its acquisition time scales with an extremely flat slope, demonstrating that our variance-based strategy is highly scalable and exceptionally well-suited for massive candidate pools.
>
> ### 4. Analysis of the Selected Samples
> **Reviewer's Request:** *Provide some analysis of the samples selected during active learning (e.g., what kinds of graphs are chosen by the variance-based strategy) to better explain why the method improves performance.*
>
> **Our Response:** This is a very insightful request. To explain *why* our graph-centric variance strategy outperforms pair-level strategies, we have added a comprehensive qualitative and quantitative analysis of the selected graphs in the new **Section 5.3.2**, **Figure 6**, and **Figure 7**.
> *   **Qualitative Analysis (Figure 6):** Visualizations confirm that EVGS selects topologically diverse graphs that act as "structural anchors." By querying instances with high uncertainty variance, EVGS identifies graphs that lie in transition/ambiguous regions of the metric space. Labeling these pivotal instances effectively calibrates the regression scale and provides comprehensive coverage of the structural manifold, explaining the superior generalization of our method.
> *   **Quantitative Analysis (Figure 7):** We investigated the topological properties (graph size and average degree) of the selected subsets. The results starkly illustrate the "hubness trap": standard pair-level strategies (EPS) are easily misled by dense, complex substructures, wasting the budget on highly redundant graphs. Conversely, EVGS selects a subset with a uniform distribution spanning the entire range of structural metrics (from sparse to dense).
>
> ***
>
> We are deeply grateful for your constructive feedback, which has undoubtedly elevated the rigor and clarity of our paper. We hope that these newly added empirical validations fully address your concerns.

---

### Review · Reviewer_kwjV · 2026-03-22

**Summary Of Contributions:**

### Summary
This paper introduces EVGS (Evidential Variance for Graph Selection), a novel active learning framework for the Graph Similarity Learning (GSL) task (which targets measuring the similarity between two graphs), designed to target the high annotation cost for training GNN-based similarity models on graph pairs. Specifically, the authors identify two challenges in active learning for GSL: (1) the difficulty of uncertainty estimation in continuous regression settings; and (2) the combinatorial dependency of paired graph inputs. After that, the authors address them by combining evidential deep learning for uncertainty estimation with a graph-centric selection strategy that measures the variance of uncertainty across the interactions of graphs. In addition to this, to further stabilize learning under low-data regimes, they propose an MSE-anchored regularization term to prevent gradient shrinkage and ensure meaningful uncertainty signals. Extensive experiments on multiple benchmarks demonstrate that EVGS consistently outperforms existing active learning baselines, achieving strong performance with fewer labeled pairs.

### Strengths
* This work formulates and targets the novel problem of active learning for graph similarity learning.
* The proposed method is well-motivated, which is grounded in challenges unique to the graph similarity learning task.
* The authors perform experiments across multiple datasets, backbones, and ablations, which demonstrates the efficacy of the proposed approach.
* This paper is well-written and easy to follow.

### Weaknesses
* While the identified challenges are sound, many of the core components (e.g., evidential regression and uncertainty-based selection) build upon existing techniques, making it somewhat unclear what the key methodological contributions are beyond their integration.
* While the selection complexity seems to be reduced from O(N^2) to O(NK), this aspect is not fully analyzed. In other words, while this improves scalability with respect to the number of graphs, it remains unclear how the method scales to large graphs with massive numbers of nodes and edges, as it still relies on pairwise GNN-based similarity computations.

**Audience:**

Yes

**Audience Explanation:**

The work addresses a novel and practically relevant setting of active learning for graph similarity learning (with implications for data-efficient learning), although the scope of the problem may be somewhat niche and limited.

**Broader Impact Concerns:**

I believe that this work does not have a direct broader impact concern.

**Claims And Evidence:**

Yes

**Claims Explanation:**

The claims are supported by consistent improvements across multiple benchmarks, backbones, and thorough ablation studies.

**Requested Changes:**

Please see my comments in the Summary Of Contributions field.

---

> ### Author Response · Authors · 2026-03-26
>
> We sincerely thank the reviewer for their positive evaluation of our work, particularly for recognizing the novelty of the problem formulation, the strong motivation grounded in GSL-specific challenges, the thoroughness of our experiments, and the clarity of our writing.
>
> Below, we address your constructive feedback point by point and detail the corresponding revisions we have made to the manuscript (highlighted in **RoyalBlue** in the revised paper).
>
> ---
>
> ### **Response to Weakness 1: Clarification of Methodological Contributions**
>
> > **Reviewer's Comment:** *While the identified challenges are sound, many of the core components (e.g., evidential regression and uncertainty-based selection) build upon existing techniques, making it somewhat unclear what the key methodological contributions are beyond their integration.*
>
> **Our Response:**
> We completely understand the reviewer's perspective. While EVGS indeed builds on established concepts such as evidential learning and uncertainty sampling, we would like to emphasize that our contribution is **not a straightforward "plug-and-play" integration**. Applying these techniques to the novel setting of Active Learning for Graph Similarity Learning (AL for GSL) exposed fundamental bottlenecks that required specific, non-trivial methodological innovations.
>
> To make our core contributions explicitly clear beyond mere integration, we have added a dedicated **Section 4.3 (Discussions)** in the revised manuscript. We highlight our innovations across three distinct perspectives:
>
> 1.  **Methodological Perspective (Adapting Evidential Regression for AL):** Standard evidential regression fails catastrophically in the low-data regime of early AL cycles due to the "Low-Evidence Trap" (vanishing gradients). Our key methodological contribution here is identifying this specific pathology and proposing a tailored **MSE-anchored regularizer** to inject gradients and prevent evidence collapse. Without this specific innovation, evidential regression cannot be used for AL in regression tasks.
> 2.  **Algorithmic Philosophy (Graph-centric vs. Pair-level):** Standard uncertainty sampling assumes instance independence. We identified that directly applying this to paired inputs leads to the "Hubness Trap" (severe structural redundancy). Our contribution is a philosophical shift: introducing a **graph-centric acquisition strategy based on uncertainty variance**. This explicitly models the structural interactions among candidate pairs, transitioning AL from isolated instance selection to holistic graph sampling.
> 3.  **Task Perspective:** We pioneer the formalization of AL specifically for GSL, bridging the gap between data-efficient learning and graph matching.
>
> **Changes in Manuscript:**
> *   Added **Section 4.3 Discussions** to explicitly articulate these three core innovations and differentiate EVGS from a simple combination of existing techniques.

---

> ### Author Response · Authors · 2026-03-26
>
> ### **Response to Weakness 2: Scalability Regarding Graph Size (Nodes/Edges)**
>
> > **Reviewer's Comment:** *While the selection complexity seems to be reduced from $O(N^2)$ to $O(NK)$, this aspect is not fully analyzed. In other words, while this improves scalability with respect to the number of graphs, it remains unclear how the method scales to large graphs with massive numbers of nodes and edges, as it still relies on pairwise GNN-based similarity computations.*
>
> **Our Response:**
> This is an excellent and insightful point. The reviewer correctly distinguishes between the scalability of the candidate pool ($N$ graphs) and that of the individual graphs (number of nodes/edges).
>
> We would like to clarify that **EVGS is a strictly model-agnostic meta-framework**.
> *   Our $O(N^2) \to O(NK)$ reduction successfully solves the *combinatorial search space bottleneck* inherent to the Active Learning process.
> *   However, the computational cost of evaluating a single pair (the node/edge-level complexity) is inherently tied to the **underlying GSL backbone architecture**, rather than the EVGS framework itself.
>
> Because EVGS is backbone-agnostic, it can smoothly scale to datasets with massive graphs. To achieve this, practitioners can equip EVGS with lightweight, embedding-based GSL models (which compute graph-level embeddings independently and then apply a simple distance metric, e.g., Cosine distance) instead of computationally heavy interaction-based models (which rely on dense, cross-graph node-to-node attention). Therefore, EVGS delegates the trade-off between inference latency and representation expressiveness entirely to the user's choice of backbone, ensuring broad applicability across any graph size.
>
> **Changes in Manuscript:**
> *   We have added a detailed discussion on this exact topic in **Section 5.3.1 (Scalability of EVGS)**. Specifically, we added a new paragraph starting with *"Third, regarding graph-size scalability, it is crucial to note that while our graph-centric selection successfully reduces the combinatorial complexity..."* to explicitly clarify how EVGS handles massive graphs via its model-agnostic design.
>
> ---
>
> We hope these clarifications and the corresponding revisions adequately address your concerns. We thank you again for your valuable feedback, which has helped us significantly improve the clarity and depth of our paper.

---

### Decision · Action_Editor_AEYQ · 2026-05-05

**Recommendation:** Accept with minor revision

**Additional Comments:**

I recommend the authors make the following minor revisions:
- The paper should more carefully distinguish between genuinely new contributions and adaptations/integrations of existing techniques. The proposed framing and graph-centric acquisition strategy are useful, but the language should avoid overstating the novelty beyond what is supported.
- The authors should explicitly state that EVGS improves scalability with respect to the number of candidate graphs through the $O(NK)$ acquisition approximation, but scalability with respect to large individual graphs remains dependent on the chosen GSL backbone and has not been empirically validated on very large graphs.
- The new evidence on evidence collapse, uncertainty-error correlation, pool-size scalability, and selected-graph properties is important for supporting the paper’s mechanisms. These results should remain clearly described in the main text or be prominently referenced from the appendix.
- The result showing that EVGS achieves the same order of magnitude as the full-data baseline is useful, but the paper should avoid implying that it fully matches supervised training performance, since there remains a nontrivial gap in absolute MSE.

**Audience:**

Yes

**Audience Explanation:**

The paper addresses a relevant and underexplored problem: reducing annotation cost in graph similarity learning via active learning. This topic is likely to interest researchers working on graph machine learning, graph similarity/search, active learning, uncertainty estimation, and data-efficient learning.

**Claims And Evidence:**

Yes

**Claims Explanation:**

The paper proposes EVGS, an active learning framework for graph similarity learning that combines evidential regression, an MSE-anchored regularizer, and a graph-centric uncertainty-variance acquisition strategy. The main claims are now supported by a reasonably comprehensive set of experiments and analyses.

The empirical evaluation covers three standard graph similarity benchmarks and two representative GSL backbones. EVGS is compared against several active learning baselines, including random selection, Coreset, clustering-based methods, BADGE, MC-Dropout-based uncertainty estimation, deep ensembles, and relevant internal variants. Across these settings, EVGS consistently improves MSE, Spearman correlation, and P@10, which supports the claim that the proposed strategy is effective for data-efficient GSL.

The revised version also addresses several earlier concerns. In particular, the authors added empirical evidence for the low-evidence/gradient-shrinkage issue by tracking the evolution of the evidence parameter, analyzed the correlation between epistemic uncertainty and prediction error, evaluated acquisition scalability under controlled pool expansion, and provided qualitative and quantitative analysis of selected graphs. These additions make the mechanistic claims more convincing than in the original submission.

---

> ### Author Response · Authors · 2026-05-13
>
> Dear Action Editor,
>
> Thank you for the positive evaluation and the constructive suggestions. We have incorporated all of them into the final manuscript. A brief summary follows.
>
> **De-anonymization and highlight removal.** The camera-ready version has been de-anonymized (author names and affiliations are now included), and all colored highlights previously used to indicate revisions in response to reviewer comments have been removed.
>
> **On distinguishing contributions from adaptations (Comment 1).** We added a dedicated Discussion section (Section 6) that explicitly separates what EVGS builds upon (e.g., evidential regression as an adopted probabilistic framework) from what it introduces (the MSE-anchored regularizer for gradient shrinkage in low-data regimes, and the uncertainty-variance-based graph-centric acquisition criterion). We also reviewed the language throughout the paper to ensure claims accurately reflect the scope of our contributions.
>
> **On scalability clarification (Comment 2).** A Limitation paragraph has been added at the end of Section 6. It states explicitly that the O(NK) approximation improves scalability with respect to the number of candidate graphs, while scalability for very large individual graphs remains dependent on the GSL backbone and has not been empirically validated.
>
> **On keeping mechanistic evidence in the main text (Comment 3).** All key empirical analyses now reside in the main text: evidence collapse and the gradient-anchoring effect (Figure 3(a), Section 5.2.2), uncertainty–error correlation (Figure 3(b), Section 5.2.2), pool-size scalability (Figure 5, Section 5.3.1), and selected-graph distributional analysis (Figures 6–7, Section 5.3.2). Only the supplementary baseline comparisons on the AIDS and LINUX datasets are placed in Appendix B, as the IMDB results in Figure 4 already demonstrate the full comparative trends; the appendix figures are explicitly referenced from Section 5.2.3 for completeness.
>
> **On avoiding overclaiming versus full-data performance (Comment 4).** We have removed any language that could be interpreted as claiming parity with full-data supervised training. All comparisons are framed in terms of relative improvements among AL strategies and data efficiency.
>
> Thank you again for your guidance in strengthening the paper.
>
>
> Best regards,
> Authors

---

> ### Comment · Action_Editor_AEYQ · 2026-05-28
>
> Per the TMLR's regulation, 16-page camera-ready version is good. Sorry for the delay.

---

> > ### Author Response · Authors · 2026-05-29
> >
> > Dear Action Editor,
> >
> > Thank you very much for the clarification regarding the 16-page limit and for confirming our camera-ready submission. We truly appreciate your help and guidance.
> >
> > We would also like to take this opportunity to express our sincere gratitude to you and all the reviewers for the time and effort dedicated to evaluating our work. The constructive feedback we received throughout the review process has been invaluable in improving the quality of our paper.
> >
> > Thank you again for your support!
> >
> > Best regards,
> >
> > Authors